# MetaboAnalystR 4.0: a unified LC-MS workflow for global metabolomics

Zhiqiang Pang[1], Lei Xu[1], Charles Viau[1], Yao Lu[2], Reza Salavati[1], Niladri Basu[1] & Jianguo Xia [1,2] ✉

The wide applications of liquid chromatography - mass spectrometry (LC-MS) in untargeted metabolomics demand an easy-to-use, comprehensive computational workflow to support efficient and reproducible data analysis. However, current tools were primarily developed to perform specific tasks in LC-MS based metabolomics data analysis. Here we introduce MetaboAnalystR 4.0 as a streamlined pipeline covering raw spectra processing, compound identification, statistical analysis, and functional interpretation. The key features of MetaboAnalystR 4.0 includes an auto-optimized feature detection and quantification algorithm for LC-MS1 spectra processing, efficient MS2 spectra deconvolution and compound identification for data-dependent or data-independent acquisition, and more accurate functional interpretation through integrated spectral annotation. Comprehensive validation studies using LC-MS1 and MS2 spectra obtained from standards mixtures, dilution series and clinical metabolomics samples have shown its excellent performance across a wide range of common tasks such as peak picking, spectral deconvolution, and compound identification with good computing efficiency. Together with its existing statistical analysis utilities, MetaboAnalystR 4.0 represents a significant step toward a unified, end-to-end workflow for LC-MS based global metabolomics in the open-source R environment.

Metabolomics involves the comprehensive identification and quantification of small compounds in biological samples using various analytical techniques[1]. Liquid chromatography - mass spectrometry (LC-MS) has been the primary analytical platform for global or untargeted metabolomics and exposomics[2,3]. Following spectra acquisition, common bioinformatics tasks include spectra processing, compound identification, statistical analysis, and functional interpretation of key patterns and significant signatures[4]. Over the past decades, powerful computational tools have been developed for individual tasks such as XCMS, MS-DIAL, MZmine, and asari for raw spectral processing[5–9]; GNPS, SIRIUS, and MS-FINDER for compound identification[10–12]; MetaboAnalyst and various R packages for statistical analysis[13,14]; KEGG and *mummichog* for functional interpretation[15–17]. Coordinating these different tools to enable streamlined data analysis and comprehensive understanding represents a major barrier to current LC-MS-based metabolomics and exposomics studies.

To enable a unified metabolomics workflow, several critical issues need to be addressed in spectra processing and interpretation. To facilitate both quantitative analysis and compound identification, LC-MS untargeted metabolomics are typically conducted with MS1 full scans coupled with tandem MS (MS2) using data-dependent acquisition (DDA) or data-independent acquisition (DIA) methods[18,19]. Although DDA usually acquires MS2 spectra by fragmenting precursor ions within a relatively narrow window, recent studies show that over half of them are chimeric and need to be deconvolved[18]. In contrast, DIA fragments all ions in a wider *m/z* range with multiple cycles to improve the metabolome coverage, and spectra deconvolution is essential to relink precursors with fragment ions. Sequential window

[1]Faculty of Agricultural and Environmental Sciences, McGill University, Ste-Anne-de-Bellevue, QC, Canada. [2]Department of Microbiology and Immunology, McGill University, Montreal, QC, Canada. ✉e-mail: jeff.xia@mcgill.ca

acquisition of all theoretical fragment ion mass spectra (SWATH-MS) is a specific variant of DIA to help overcome DDA drawbacks by leveraging all ions[20]. Despite multiple tools and algorithms[12,18,21–24], integrated LC-MS1 and MS2 spectra processing remains a primary bottleneck in terms of computational efficiency as well as the requirement of deep knowledge for parameter settings associated with using these tools. Most of them can only process either DDA or DIA data. There is a need for a highly efficient tool capable of supporting both methods. Secondly, MS2-based compound identification relies heavily on the availability of reference spectra databases. The lack of a comprehensive yet easily customizable reference spectra database remains an essential gap in current global metabolomics. Thirdly, compound identifications typically involve matching *m/z* values and retention times of MS1 features, as well as their associated MS2 patterns against reference databases. This process often yields multiple candidates and requires further time-consuming manual curation before functional interpretation can be performed. Traditional functional enrichment analysis relies on accurate compound identification. Recent studies have shown that, despite uncertainties at individual compound levels, functional activities can be reliably predicted by leveraging the patterns of the putative identifications based on *m/z* values and retention times of MS1 features[15,16]. The performance could be further improved by utilizing MS2 spectral data. To meet these emerging demands from the growing applications of metabolomics across broad areas, we have developed MetaboAnalystR version 4.0 to integrate LC-MS1 and MS2 spectra processing, compound identification, statistical analysis, as well as functional interpretation within a unified software environment.

MetaboAnalystR 1.0 was released in 2018 as the underlying R package of the popular MetaboAnalyst web server to facilitate transparent, reproducible statistical data analysis. The subsequent releases (version 2.0 in 2019 and version 3.0 in 2020) added important capacities for LC-MS1 spectra processing and functional interpretation, respectively. The R package allows easy creation of flexible data processing pipelines that complement the pre-defined workflows by the MetaboAnalyst web server. It is well received by the metabolomics community as evidenced by >1000 citations[25–27].

Here, we introduce MetaboAnalystR 4.0 developed as a unified computational workflow for LC-MS-based global metabolomics. In addition to enhancing and consolidating the LC-MS1 spectral processing and associated statistics and functional analysis modules, MetaboAnalystR 4.0 contains several key features including (1) an auto-optimized DDA data deconvolution workflow to deal with chimeric spectra; (2) an efficient SWATH-DIA data deconvolution pipeline; (3) a comprehensive collection of reference spectra databases (>1.5 million spectra) coupled with common search algorithms supporting custom plug-in databases, and (4) more accurate functional activity prediction by integrating LC-MS1 and MS2 results. To better understand the performance characteristics of both individual components as well as the complete workflow, we performed comprehensive validation studies using seven different datasets ranging from standards mixtures, dilution series, exposomics data to clinical samples.

## Results

### General workflow
MetaboAnalystR 4.0 features end-to-end support for metabolomics data analysis, spanning from raw LC-MS spectra processing to statistical and functional analysis. In addition to generic data tables, compound or LC-MS peak lists, MetaboAnalystR accepts raw spectra in open formats (mzML, mzXML, mzData, netCDF and MGF) from LC-MS metabolomics experiments, as well as output files from other well-established spectra processing tools, including MS-DIAL, MZmine, XCMS, asari, MS-FINDER, and SIRIUS. The pre-processed spectral peaks and identified compounds will be automatically formatted and enter a unified pipeline for downstream processing, statistics, and functional analysis as shown in Fig. 1.

### LC-MS spectra processing
MetaboAnalystR features an auto-optimized LC-MS1 spectra processing pipeline since version 3.0. The workflow involves extracting regions of interest followed by parameter optimization based on the design of experiments[27]. The optimized parameters are subsequently used for peak detection, quantification, alignment, and annotation. This approach achieves good performance with high computing efficiency. In version 4.0, the LC-MS spectra pre-processing workflow has been significantly expanded to support integrated analysis of MS1 and MS2 spectra data (DDA or SWATH-DIA).

For DDA data, MetaboAnalystR first assigns all MS2 spectra from a single sample into different feature groups based on the mass-to-charge ratio (*m/z*) and retention time of their precursor ions. The chimeric status is evaluated based on the nearest MS scans. Briefly, MS2 spectra of all ions (including the main precursor and other contaminating ions within the isolation window and above the intensity threshold) are extracted from MS2 spectral reference libraries as candidate spectra ("Methods" section, Fig. 2a). If any reference spectrum is missing, a predicted spectrum will be generated using a similarity-network model[28] ("Methods" section, Fig. 2b). All candidates are then used to obtain the deconvolved spectrum based on a

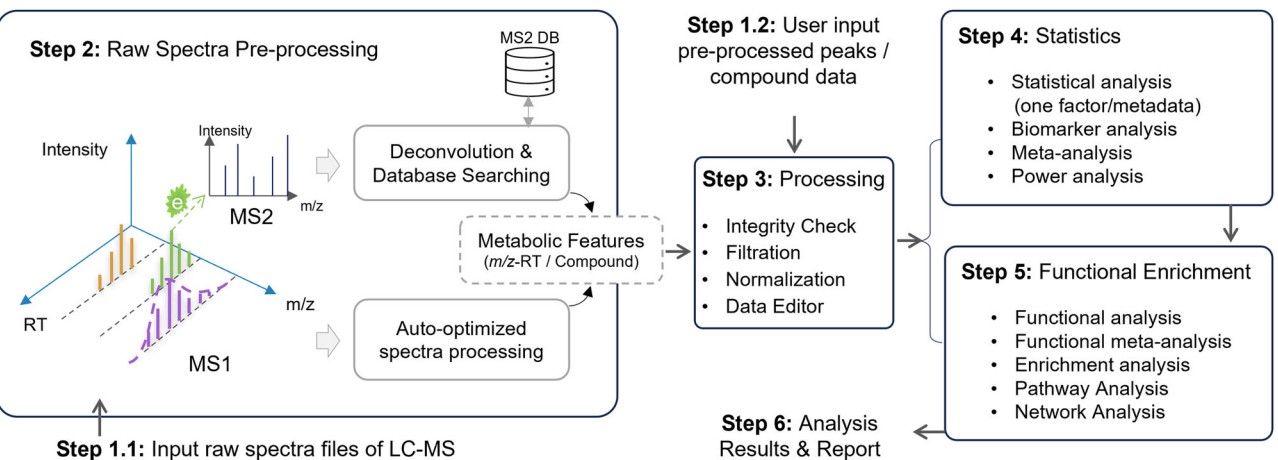

**Fig. 1 | LC-MS untargeted metabolomics workflow based on MetaboAnalystR 4.0.** Mass spectra are processed separately for MS1 and MS2 levels based on its auto-optimized workflows. MetaboAnalystR 4.0 also accepts pre-processed results from other spectra processing tools. The resulting metabolite feature table will be filtered and/or normalized for statistical and functional analysis.

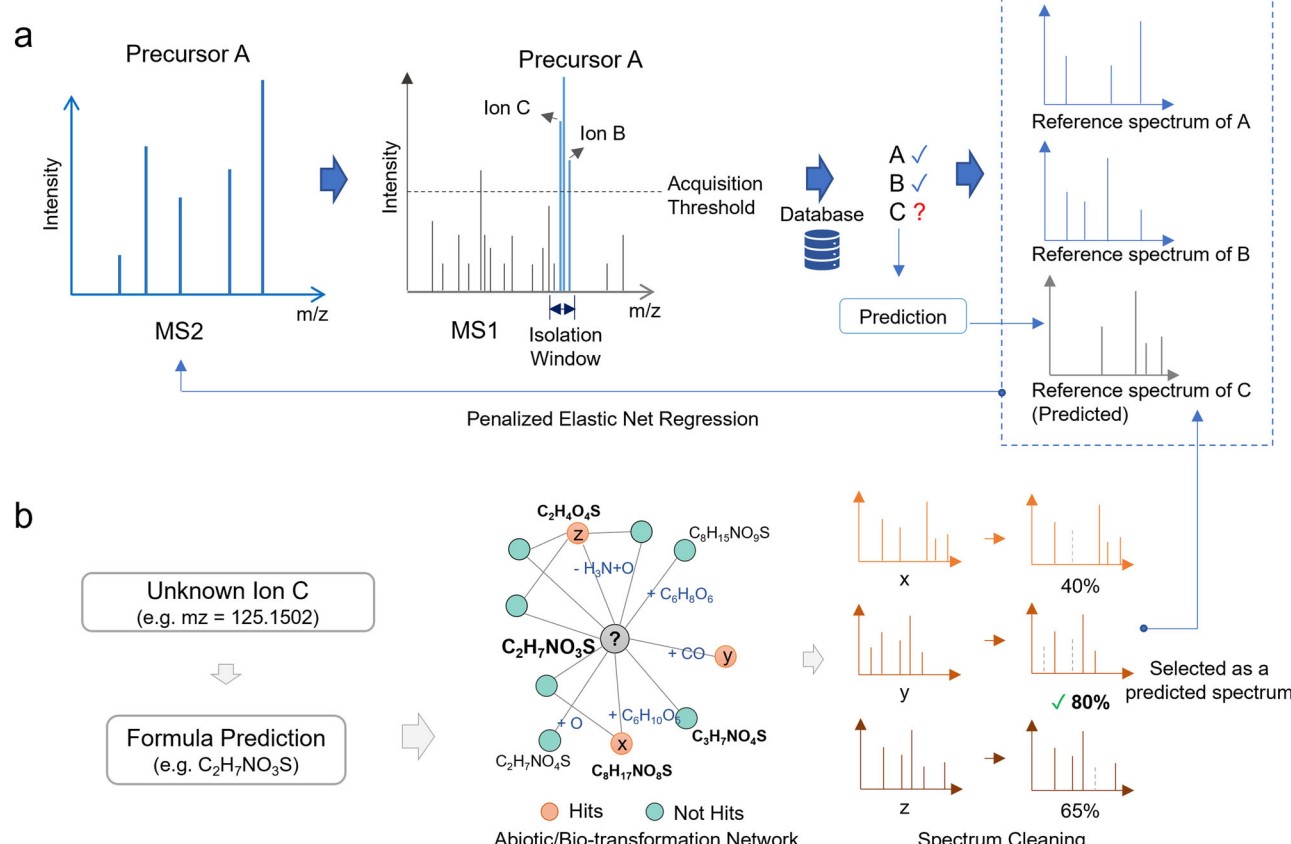

**Fig. 2 | DDA spectra deconvolution in MetaboAnalystR 4.0. a** The overall workflow. All spectra are evaluated as "clean" or "chimeric" based on the MS signals of the nearest MS1 scan. The reference spectra of all ions (A, B, and C) within the isolation window and above the intensity threshold are extracted from a reference MS2 library for regression analysis. **b** Diagram of reference spectrum prediction strategy. MetaboAnalystR can predict one or more candidates missing in the MS2 library. Formula(s) of the ion (e.g., ion C) is predicted at first. An abiotic/bio-transformation network is constructed for the formula (e.g., $C_2H_7NO_3S$), and all neighbors with reference spectra of the formula are extracted as a list. Each fragment of the single spectrum in the list is predicted in a formula. If the formula (e.g., $C_3H_9$) of the fragment includes more chemical elements (number or type) than the original formula (e.g., $C_2H_7NO_3S$), it is excluded from the spectrum. The clean spectrum is returned to the list. Their similarities to the original MS2 spectrum of all spectra in the list are evaluated, and the one with the highest similarity score is selected as the predicted spectrum for the ion.

self-tuned regression algorithm ("Methods" section). The SWATH-DIA data processing is based on the DecoMetDIA approach[23], with the core algorithm re-implemented using the Rcpp/C++ framework and further optimized to support parallel computing to address the computational bottleneck associated with the original implementation.

Following the deconvolution step, all MS2 spectra from replicates (if any) for a specific MS feature undergo a consensus step to generate a single spectrum to minimize potential errors and noise. This process may optionally utilize the observed frequency of a specific fragment. For instance, if a fragment is detected only once in replicates and is not seen in any reference MS2 spectra, it may be optionally excluded ("Methods" section, Supplementary Fig. 1).

The consensus spectra are searched against a reference spectra database for compound identification ("Methods" section). Users can choose different databases based on instrument types, collision energies, and other common options. Two widely used methods – dot product and spectral entropy have been implemented to evaluate MS2 matching similarity[29]. The candidates for a particular feature are scored by considering *m/z*, retention time, isotope, and MS2 similarity score together[5]. The matching scores range between 0 and 100, where 0 indicates no matching and 100 indicates a perfect match ("Methods" section). The top N candidates, defined by users, can be exported as the database search results. If the matching score is below 10, MetaboAnalystR optionally performs a neutral loss scan to further improve the compound identification rate[30].

To be interoperable with other widely used tools, MetaboAnalystR accepts LC-MS1 feature detection results from MS-DIAL, MZmine, asari, and XCMS[5–9]. The results can be automatically formatted into the generic format for processing by MetaboAnalystR. Additionally, MetaboAnalystR accepts the pre-processed LC-MS2 results in MSP or MGF files generated by MS-DIAL and MZmine. Users could input these files directly for MS2 database search and export the results for downstream analysis. MetaboAnalystR also offers functionalities to convert the compound identification results from MS-FINDER and SIRIUS for functional analysis[10–12].

**Comprehensive MS2 spectra database**

MetaboAnalystR 4.0 comes with a comprehensive reference spectra database organized under five different themes or libraries, including pathway compound library, biology compound library, lipid library, exposome library, and the complete library. A summary of these five libraries is presented in Table 1. These databases have been curated from public repositories, including HMDB[31], MoNA Series[32], LipidBlast[33], MassBank[32], GNPS[22], LipidBank[34], MINEs[35], LipidMAPs[36], and KEGG[17] ("Methods" section). All fragments in the MS2 spectra reference libraries have been further annotated into formulas using BUDDY[37]. In addition, the corresponding neutral loss spectra databases have also been pre-calculated, corresponding to the five databases above. The underlying spectra, which include original MS2 spectra, neutral loss, and annotated fragments, are stored as

**Table 1 | Summary of MS2 reference spectra libraries curated from different public resources**

| Name | Records | Unique compounds | Size (MS2 \| neutral loss) |
|---|---|---|---|
| Complete library | 10,420,215 | 1,551,012 | 7.2 \| 6.4 GB |
| Pathway library | 172,370 | 3456 | 138.2 \| 94.1 MB |
| Biology library | 864,386 | 49,055 | 744.0 \| 491.0 MB |
| Exposome library | 1,883,828 | 106,387 | 1.5 \| 1.1 GB |
| Lipid library | 3,221,409 | 878,220 | 1.9 \| 1.1 GB |

SQLite files available for download. The simple schema of the SQLite database allows users to easily prepare their own in-house custom databases for compound searching.

## Integrating LC-MS data processing results for accurate functional insights

Obtaining functional insights is among the main interests in metabolomics studies. Conventional approaches generally require manual annotation of a significant portion of spectral features, which is a very time-consuming process. The *mummichog* algorithm[15] has demonstrated that putatively annotated compounds from high-resolution MS peaks are able to collectively point to the correct pathways. A pivotal step in this process is provisionally mapping $m/z$ information to empirical compounds. The initial algorithm (version 1.0) uses $m/z$ values of MS peaks. The subsequent release (version 2.0) incorporates peak retention times. MetaboAnalystR 4.0 further utilizes MS2 information to filter out impractical assignments to improve accuracy and specificity. After MS1 and MS2 spectral processing, MetaboAnalystR can automatically perform statistical analysis from the peak intensity table and format the database searching results for functional enrichment analysis ("Methods" section). The default database for functional analysis is based on the known pathways curated from KEGG and BioCyc[38]. Users can also choose other functional libraries based on disease signatures, chemical ontologies, SNP-associated metabolites, etc[39].

## Benchmarking and case studies

Using a total of seven datasets, including one whole blood exposomics dataset, one serial dilution dataset, three standards mixtures datasets[18,40,41], and two clinical plasma metabolomics datasets[42,43], we evaluated the performance of MetaboAnalystR 4.0 for LC-MS1 spectra processing and quantification, MS2 spectra processing and compound identification, statistical analysis, and functional interpretation. When possible, we also compared with those obtained using other commonly used tools, and the results were provided as supplementary materials.

## Performance of LC-MS1 spectral processing and quantification.
Understanding the chemical compositions of different types of blood samples is important, especially for diseases whose pathogenesis is closely related to blood cellular components[44,45]. To evaluate the performance of LC-MS1 spectral processing workflow, we have generated a whole blood exposomics dataset to compare the differences in plasma, serum, and whole blood from 14 individuals using LC-MS-based global metabolomics. To ensure comprehensive metabolome coverage, LC-MS1 spectra were collected using two column types under positive and negative ion modes (C18-ESI⁺, C18-ESI⁻, HILIC-ESI⁺, and HILIC-ESI⁻) for each sample. MS2 has been acquired in both DDA and SWATH-DIA methods. The general design of this study is shown in Fig. 3a.

All MS features were detected with the auto-optimized LC-MS1 workflow in MetaboAnalystR. Over 12,000 MS1 features have been detected for each individual mode (Fig. 3b). Statistical analysis was performed on the feature abundance tables. Principal component analysis (PCA) shows significant differences in the metabolomes of the three different types of blood samples (Fig. 3c). A clustering heatmap was used to view abundance distributions of all MS1 features (Fig. 3d). Generally, whole blood contains more metabolite features compared to serum and plasma. Serum does not metabolically equate to plasma as shown by the presence of unique features in both data types. In this study, we focused on elucidating the distinct features among different types of blood samples. Over 2000 unique metabolic features have been found in this dataset (Fig. 3b). Similar results have also been observed from the other three datasets (Supplementary Figs. 2–6).

To further evaluate the quantification performance of MetaboAnalystR, a dilution series ($n = 13$) was prepared by mixing serum and urine samples with varying ratios in a cross-gradient manner[7] (Fig. 3e, "Methods" section). The cross-gradient dilution is designed to minimize the potential bias caused by matrix effects in equal dilutions[46]. We conducted a Pearson correlation analysis between the dilution ratios and the MS1 feature intensities obtained using MetaboAnalystR. As illustrated in Fig. 3f, over 90% of MS1 features detected by MetaboAnalystR exhibited high average correlation coefficients (>0.85), indicating its excellent quantification performance. Compared to other tools under their default settings, MetaboAnalystR exhibits the overall best quantification results (Supplementary Figs. 7–9).

**Performance of LC-MS2 spectral processing.** Three standards mixtures (SM) datasets with different levels of complexities were downloaded from Mass Spectrometry Metabolite Library of Standards (MSMLS, IROA Technologies). The 1st SM dataset contains a mixture of 526 standards (complex mixture), including one DDA injection and one SWATH-DIA injection (single replicate for both ESI⁺ and ESI⁻). The 2nd SM includes 15 DDA samples, each containing 10–15 non-isobaric compounds (simple mixture, single replicate for both ESI⁺ and ESI⁻). The 3rd SM dataset contains 91 compounds (medium complexity) and was analyzed by both DDA and SWATH-DIA (three replicates each for both ESI⁺ and ESI⁻)[40]. The chemical identity information for each compound (retention time, m/z, and InChIKey) in each SM dataset was utilized as the corresponding reference compound lists to assess the accuracy of compound identification.

The 1st SM dataset was used to evaluate the performance of DDA deconvolution. The results indicate that performing deconvolution on DDA spectra has increased the number of compounds that were correctly identified as the top candidates (Table 2, Supplementary Tables 1–3, Supplementary Fig. 10), and the matching scores of the identified compounds reported by the deconvolution pipeline were significantly improved compared to those obtained from the pipeline without deconvolution (Fig. 4a). Compared to other tools under their default settings, MetaboAnalystR detected the highest number of compounds with high computational efficiency (Table 2). Similar results were also observed from the 2nd SM dataset, with MetaboAnalystR showing a higher compound identification rate than other tools under their default settings (Fig. 4b, Supplementary Fig. 11). While manual parameter optimization by expert users may improve the performance of other tools, the MS2 spectra processing in MetaboAnalystR is auto-optimized ("Methods" section) to avoid such requirements.

Using the 3rd SM dataset, we first tested the number of compounds that could be identified correctly. For DDA dataset, MetaboAnalystR correctly identified most compounds by using either Complete library or Pathway library (Fig. 4c). To evaluate the false discovery rate, we generated a series of decoy spectra by randomly increasing $m/z$ error and replacing the original MS2 spectra with synthetic spectra from isobaric compounds ("Methods" section, Supplementary Fig. 12). The results showed that MetaboAnalystR was

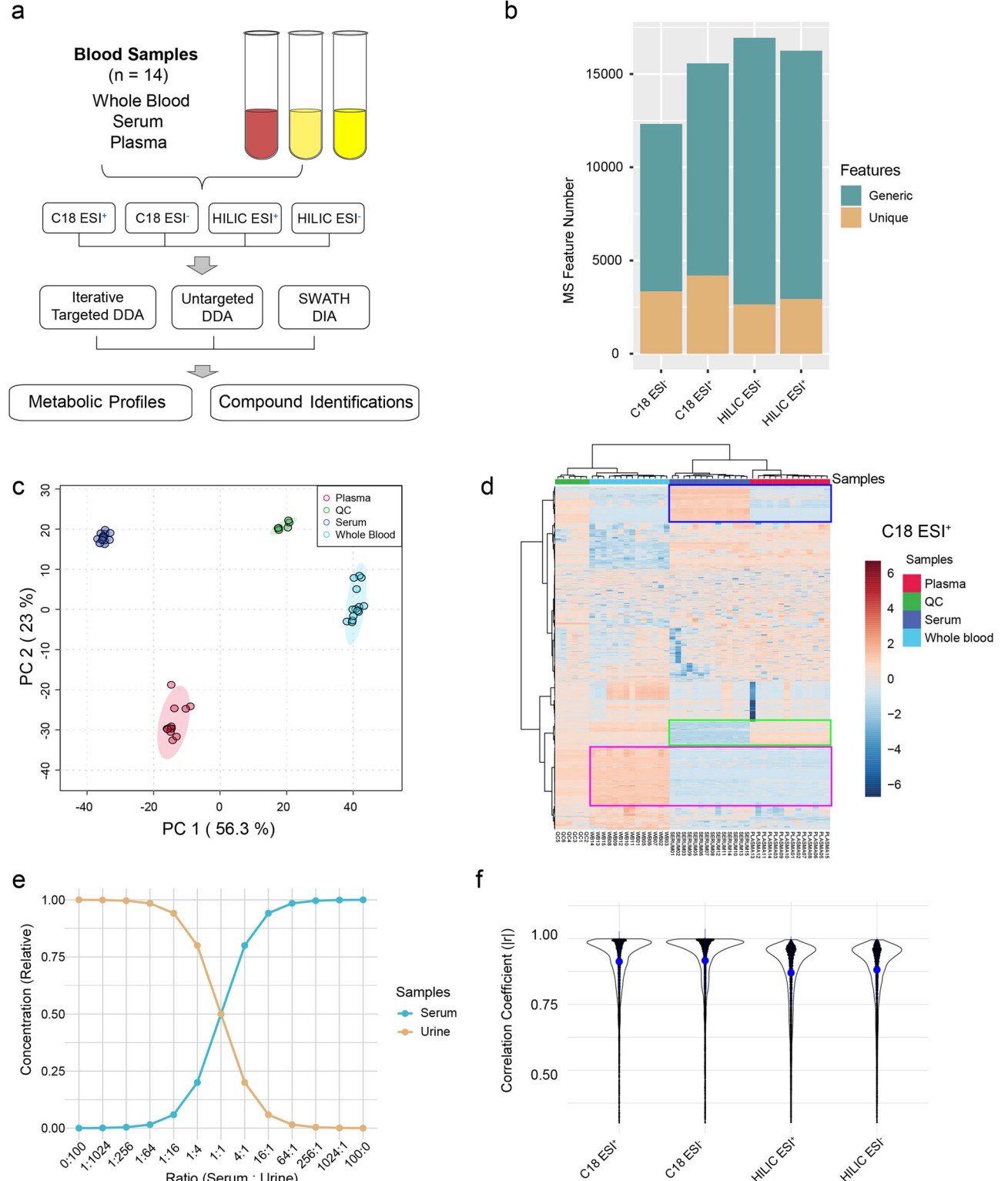

**Fig. 3 | Evaluation of LC-MS1 data processing and quantification. a** Design of the metabolomics experiment. **b** Statistics of LC-MS1 features across different modes. Generic features, MS features detected across all blood samples; Unique features, features detected only in one type of blood samples. **c** PCA score plot of the blood dataset (C18, ESI⁺). **d** Heatmap of the complete metabolic profiles (MS1 level, C18, ESI⁺). Unique MS1 features for a specific blood sample type were highlighted with rectangles. Blue, unique features for serum compared to plasma; Green, unique features for plasma compared to serum. Ruby, unique features for whole blood compared to plasma and serum. **e** Design of serial dilutions. Urine and serum are mixed according to the ratio labeled at the x-axis. **f** Correlation analysis of MS features from serial dilutions. MetaboAnalystR reported high average correlation coefficients (>0.85) across different modes, compared to other tools (Supplementary Fig. 7).

**Table 2 | Summary of identified compounds from a standards mixture of 406 compounds by different tools**

| Tools | [a]Detected (MS1) | [b]Annotated (MS2) | Percentage (%) |
|---|---|---|---|
| MetaboAnalystR (deco) | 239 | 159 | 39.2 |
| MetaboAnalystR (non-deco) | 239 | 146 | 36.0 |
| MS-DIAL/MS-FINDER | 165 | 82 | 20.2 |
| MZmine/SIRIUS | 221 | 77 | 19.0 |

The result is based on MS2 (DDA, ESI[+]). The results from the other three modes are available in Supplementary Tables 1–3.
[a]Detected: number of MS1 peaks matched the *m/z* and retention time of the standards.
[b]Annotated: number of compounds correctly identified as the top hit from the MS2 spectra search.

able to maintain a low false positive rate (<25%, Supplementary Figs. 13 and 14).

Finally, we evaluated the compound identification performance on complex biological samples using the blood exposomics dataset. Herein, we focused on the unique features of different types of blood samples. These features were used as target lists for MS2-based compound identification. As depicted in Fig. 4d, MetaboAnalystR identified over 500 compounds (level 2a) through DDA, which was ~5% more compared to compound identification based on non-deconvoluted DDA spectra (Supplementary Fig. 15). For instance, 2-Piperidinone (Fig. 4e) was only identified after deconvolution, with a matching score of 92. Over 1000 compounds were identified based on SWATH-DIA. The chemical composition analysis showed that whole blood samples contained more lipids, organic acids, and organic heterocyclic components in contrast to serum and plasma. The main chemical differences between serum and plasma are lipids (Supplementary Fig. 16).

**Evaluating the complete workflow using clinical metabolomics data.** We used two clinical metabolomics datasets collected from two recent COVID-19 studies[42,43] to validate the complete workflow of MetaboAnalystR including raw LC-MS1 and MS2 spectra processing, compound identification, statistical analysis, and biological interpretation. The 1st dataset includes 160 samples for both polar metabolites and non-polar lipids datasets (ESI[+] and ESI[−], DDA, "Methods" section) categorized into five groups (Asymptomatic, Mild, Severe, Critical, and Fatal)[42]. The 2nd dataset includes 30 samples (ESI[+] and ESI[−], SWATH-DIA, "Methods" section) categorized into disease and healthy control[43].

As shown in Fig. 5a, MetaboAnalystR detected over 5000 MS1 features for all datasets. Compound identification result indicates that MetaboAnalystR could identify around 10%-24% of compounds (level 2a, Fig. 5b), which is higher than other tools under their default settings (Supplementary Figs. 17–19). Note the comparison with other workflows or tools could only be performed till this step.

We applied the enhanced *mummichog* algorithm using the results obtained above for function analysis. We chose to compare biological differences between the Mild and Fatal groups for the 1st dataset, and between the disease and healthy controls groups for the 2nd dataset. There are two sub-datasets (polar and non-polar lipids) in the 1st dataset and one dataset in the 2nd dataset. Therefore, a total of three comparisons were performed. In general, more pathways were reported when MS2 data were used (Fig. 5c, d). Compared to the result based only on MS1 features, four more significant pathways were reported when considering the SWATH-DIA data (Fig. 5e). These pathways are related to phosphatidylinositol phosphate, vitamin D, vitamin C metabolism, and arachidonic cascades (Supplementary Tables 4–6). They were reported to be related to the pathogenesis of the disease by the previous studies[47–50].

All predicted active pathways (including those based on other tools) are summarized in Supplementary Figs. 20–24.

**Assessment of computational performance.** The two clinical metabolomics datasets were used to benchmark the computational performance of all tools included in this study. The assessments were conducted using a workstation (Dell OptiPlex 7070, 64GB RAM, Intel-i7-9700 CPU, Ubuntu 20.04.2) controlled by Simple Linux Utility for Resource Management (SLURM[51], "Methods" section). MetaboAnalystR completed raw LC-MS1 spectral processing within 5 h and MS2 spectral processing within 16 h for each of the three datasets, with memory usage below 25GB. Compared to other tools under their default settings, MetaboAnalystR achieved MS feature detection at a speed similar to MZmine and MS-DIAL (Supplementary Table 7 and Supplementary Figs. 25–26). The memory consumption of MetaboAnalystR is comparable to that of MS-DIAL but lower than that of other tools. According to the execution logs, MS2 spectra searching in SIRIUS is based on application programming interface (API) calls to remote web services, which is highly dependent on the network traffic and responsiveness of the remote server. MS-FINDER also partially uses remote access to predict formulas. In comparison, MetaboAnalystR allows users to run the complete workflow locally without depending on network traffic or remote server load.

## Discussion

Despite significant progress made in the past decades, analyzing LC-MS data from untargeted metabolomics remains challenging. Researchers have to learn to use multiple tools and R packages in order to achieve comprehensive data analysis and understanding. In addition, better support for MS2 spectra processing is required to enable more accurate, high-throughput compound identifications and functional understanding. We have developed MetaboAnalystR 4.0 to help orchestrate LC-MS spectra processing, compound identification, statistical analysis, and biological interpretation within an open-source R environment. MetaboAnalystR accepts LC-MS1 and MS2 data in common open formats. Compound identification can be performed efficiently within the same workflow using the comprehensive reference spectra libraries. The results can be directly fed into the downstream statistical and functional analysis. Our comprehensive case studies have indicated that MetaboAnalystR 4.0 performs favorably when compared with other widely used tools in terms of spectral processing, compound identification, quantification, interpretation as well as computing efficiency.

A key innovation and advantage of using MetaboAnalystR 4.0 for raw spectral processing lies in its auto-optimized design for common LC-MS1 and MS2 spectra processing. Another important contribution is the manual curation and annotation of comprehensive MS2 spectra libraries coupled with efficient search algorithms for compound identification. Users can visually explore the MS2 spectra searching results through interactive mirror plots. As an open-source R package, MetaboAnalystR is designed to be extensible and interoperable with other tools and workflows. It not only accepts common data formats (such as csv and txt) but also supports standard formats (such as mzTab). MetaboAnalystR can also directly incorporate raw spectral processing results (such as MSP and MGF files) generated from other well-established tools into its workflow for downstream analysis. Finally, MetaboAnalystR 4.0 is the backend of the MetaboAnalyst 6.0 web application[52], which has been developed to support users who are not comfortable with the R command-line interface.

An important limitation of the current implementation of MetaboAnalystR 4.0 is that the built-in database of MS2 reference spectra were compiled from various instruments and experimental conditions, resulting in potentially varying qualities in compound identification. In

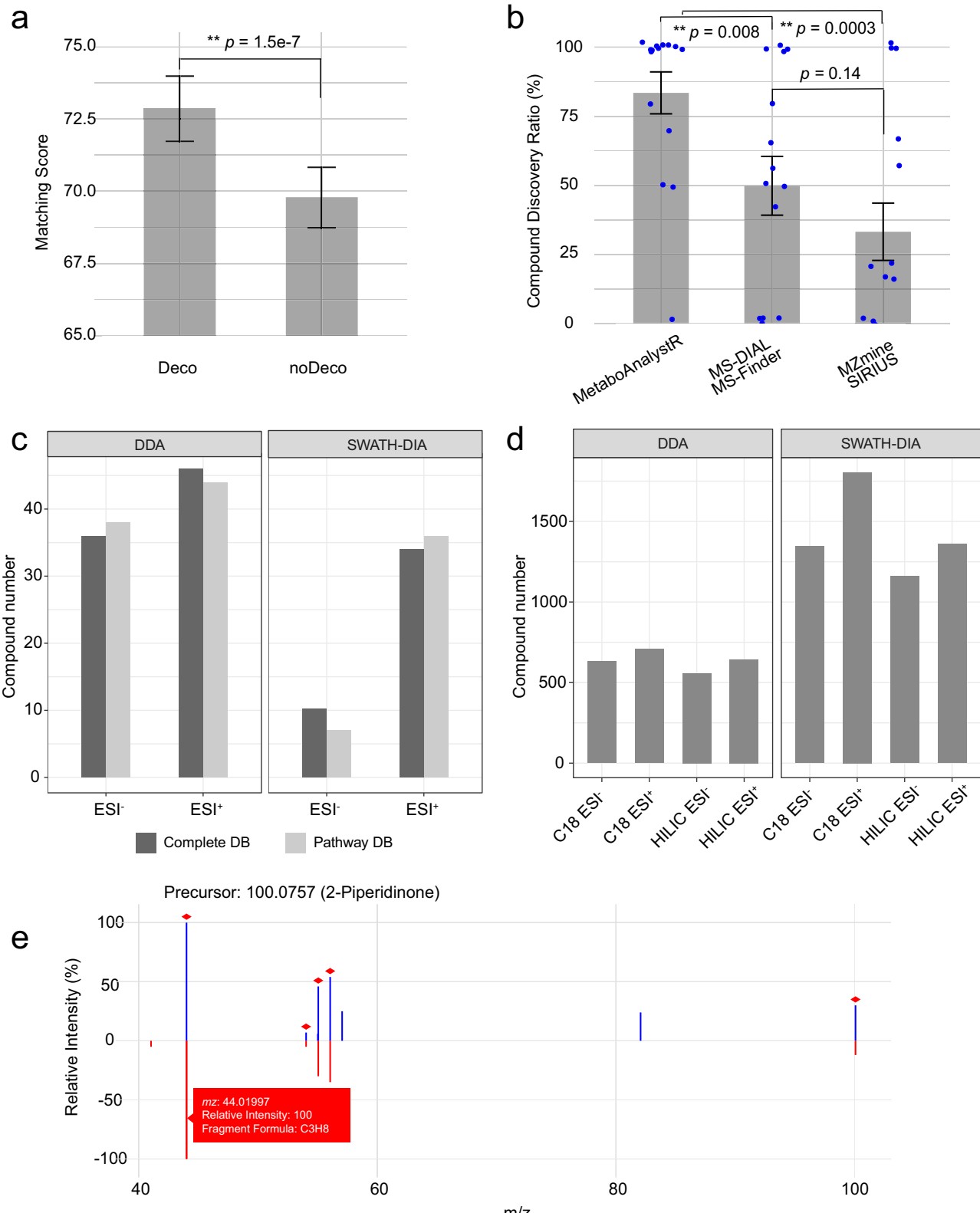

contrast, MS2 reference spectra from METLIN[53] or other commercial tools originate from in-house experimentally validated databases under more standardized conditions. We encourage users to convert their own in-house databases into the compatible SQLite format for the MS2 search workflow.

We will continue to improve the MetaboAnalystR package and the MetaboAnalyst web application to make LC-MS global metabolomics more accessible to the community. For future directions, we intend to add a new component to support the analysis of data from stable isotope-labeling experiments[54,55]. MetaboAnalystR does not support ion-mobility spectrometry and all-ion fragmentation (AIF) data[56]. They have also been increasingly applied in metabolomics studies. The support for these features will be achieved in the future release.

**Fig. 4 | Evaluation of LC-MS2 data processing and result visualization.**
**a** Comparison of matching scores of DDA (with or without deconvolution, ESI⁺) in the complex standards mixture sample. The deconvolution algorithm could significantly improve the matching scores of chemical candidates in comparison to the non-deconvolved spectra. $n = 99$ independent compounds, means ± SEM; paired one-tailed Student's $t$-test (**, significant, $p = 1.5e\text{-}7$, no adjustment).
**b** Statistical analysis of the compound identification results of the 2nd standards mixture dataset. Compared to the other two workflows, MetaboAnalystR reported a significantly higher compound identification percentage. $n = 15$ independent datasets, means ± SEM; unpaired one-tailed Student's $t$-test without adjustment (MetaboAnalystR vs. MS-DIAL/MS-Finder, significant, $p = 0.008$; MetaboAnalystR

vs. MZmine/SIRIUS, significant, $p = 0.0003$; MS-DIAL/MS-Finder vs. MZmine/SIRIUS, insignificant, $p = 0.14$). **c** Number of identified compounds from the 3rd standards mixture dataset using different MS2 reference libraries. **d** Results of MS2 spectra-based compound identification for the blood exposomics dataset. **e** Example mirror plot of 2-Piperidinone illustrating the MS2 spectrum matching pattern. The upper side (blue) represents the user's input, while the bottom side (red) displays the spectrum from the reference library. When the mouse hovers over the fragments, corresponding information, including $m/z$, relative intensity, and potential formula, is displayed. All matched fragments are marked with a red diamond at the top.

In conclusion, MetaboAnalystR 4.0 has significantly enhanced a series of important functions and implemented the much-requested support for MS2 spectra processing and compound identification in an efficient manner. In combination with its already well-established functionalities on statistical analysis and functional interpretation, MetaboAnalystR 4.0 offers a unified workflow to support LC-MS global metabolomics and exposomics studies.

## Methods
This research complies with all relevant ethical regulations. The whole blood exposomics study was approved by the Research Ethics Office of McGill University (Study ID: A05-M26-16B).

### Chemicals
Standard human serum and ammonium acetate ($NH_4AC$) were purchased from Sigma-Aldrich (Sigma, St. Louis, MO, USA). Acetonitrile (ACN), methanol (MeOH), 0.1% formic acid (FA) in Water, 0.1% FA in ACN, and pure water were purchased from Fisher Chemical (Morris Plains, NJ, U.S.A.).

### Blood sample preparation
Healthy volunteers were recruited from McGill University as previously discussed[57]. About five milliliters of venous whole blood (WB) were drawn from each volunteer into a BD K2-EDTA Trace Element free Vacutainer. A sub-sample of this whole blood was used to obtain plasma (i.e., whole blood centrifuged for 15 min at 4 °C at 2700 rpm). From each individual, another blood sample was collected into a BD Vacutainer tube not containing any anticoagulant, which was allowed to sit for ~30–60 min for clots to form following which serum was obtained by centrifugation (15 min at 4 °C at 2700 rpm). Blood samples from 14 individuals were collected and included in this study. All blood samples were paired with three different types (whole blood, serum, and plasma). All samples were immediately frozen at −80 °C until analysis. The demographic information of all subjects is summarized in Supplementary Table 8. This study was approved by the Research Ethics Office of McGill University (Study ID: A05-M26-16B).

The different blood sample types were prepared based on the previously published protocols[58,59]. The three blood sample types (WB, serum, and plasma) were thawed on ice for one hour, and then vortexed for 30 s to ensure homogeneity. One hundred μl of each sample type was transferred to a 1.5 ml Eppendorf microcentrifuge tube, to which 400 μl of −20 °C 1:1 ACN:MeOH (v/v) was added. Samples were vortexed for 60 s and stored at −20 °C for one hour. Samples were then centrifuged at 16,100 × $g$ for 10 min at 4 °C. The supernatants were collected (250 μl) and filtered by centrifugation using 0.2 μm Nanosep centrifugal filters (PALL Life Sciences) at 14,000 × $g$ for 15 min at 4 °C. Filtered samples (120 μl) were then transferred to LC-MS vials equipped with 250 μl glass inserts and run in the LC-MS. A Quality Control (QC) sample was made by pooling equal volumes from each filtered samples supernatant into one 1.5 ml Eppendorf microcentrifuge tube.

### Sample preparation for serial dilutions
A urine sample was collected from a donor from the population mentioned above[57]. A total of 100 mL urine was sub-sampled and frozen at −80 °C. A human standard serum sample (Sigma-Aldrich, Sigma, St. Louis, MO, USA) and the urine were thawed on ice for one hour, and then vortexed for 30 s to ensure homogeneity. A total of 13 Eppendorf microcentrifuge tubes were prepared and labeled from A to M. For tubes A to E, 150 μl urine was transferred into each of them. For tubes G to K, 150 μl standard serum was transferred into each of them. 250 μl pure urine was transferred into tube L, and 250 μl pure serum was transferred into tube M. Then, 50 μl pure serum were extracted and mixed into tube E. Then, 50 μl liquids were extracted and mixed into tube D, and so on until tube A. Same operations were repeated for pure urine tube L, and tube G–K. Finally, 75 μl pure urine and serum were extracted respectively and mixed into tube F. As a result, a total of 11 dilution mixtures and two pure samples were generated. The whole preparation workflow is shown in Supplementary Fig. 27. After the preparation of the serial dilutions, all samples were processed similarly as the blood samples, except that the ratio of organic reagents to samples is 1:2.5 instead of 1:4. All processed samples (120 μl) were then transferred to LC-MS vials equipped with 250 μl glass inserts and run in the LC-MS. No QC samples were prepared in this case study. Three replicates were prepared for each of the serial dilutions. This study has been approved by the Research Ethics Office of McGill University as described above.

### LC-MS and MS2 analysis
Metabolic profiling at the MS1 level was performed on a UHPLC system (Thermo Scientific™ UltiMate™ 3000 System). A hydrophobic column (Hypersil GOLD™ aQ C18 Polar Endcapped HPLC Column, 100 mm × 2.1 mm, 1.9 μm) and a hydrophilic (Accucore™ 150 Amide HILIC HPLC Column, 100 mm × 2.1 mm, 2.5 μm) column were used for reverse phase (C18 column) and hydrophilic interaction liquid chromatography (HILIC column) separation, respectively. The chromatogram system was coupled to a Thermo Scientific Q-Exactive Orbitrap mass spectrometer.

The chromatographic conditions for the C18 and HILIC columns were optimized as follows. For both columns, the flow rate was fixed at 0.4 mL/min. For the C18 columns, the composition of the mobile phases A and B was 0.1% FA in water and 0.1% FA in ACN, respectively. For HILIC chromatography, the composition of the mobile phases A and B were 50% ACN in water with 5 mmol/L $NH_4AC$ and 95% ACN in Water with 5 mmol/L $NH_4AC$, respectively. The gradient procedures and other instrumental parameters are provided in Supplementary Table 9.

The Q-Exactive Orbitrap MS was configured as follows. For the C18 column, an electrospray ion (ESI) source with a spray voltage of 4 keV in positive mode and 3.5 keV in negative mode were used, and for HILIC a voltage of 4 keV in positive mode and 3.8 keV in negative mode were used. Additional MS parameters were set for the C18 and HILIC columns, which are summarized in Supplementary Table 10. Both

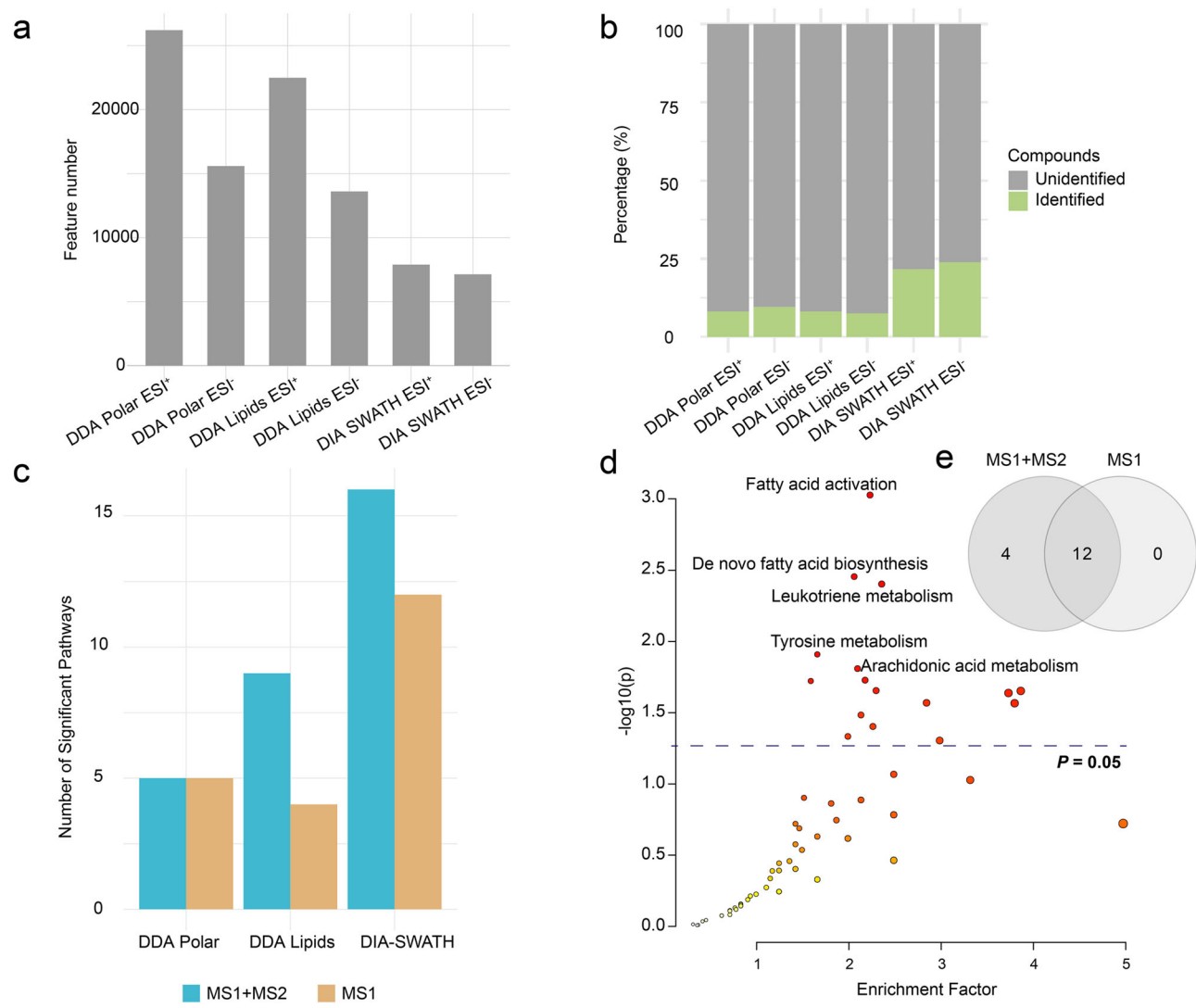

**Fig. 5 | Biological interpretation of clinical metabolomics data. a** Summary of MS1 features detected from DDA and SWATH-DIA datasets. **b** Percentage of MS1 features identified based on MS2 spectra. **c** Comparison of significant pathways obtained by integrating MS2 identification results (MS1 + MS2) or not (MS1).

**d** Scatter plot of pathway enrichment analysis results of SWATH-DIA dataset. $n = 30$ independent experiment samples. Fisher's exact test without adjustment for functional analysis. **e** Venn diagram of pathway analysis results from SWATH-DIA datasets.

positive (ESI⁺) and negative (ESI⁻) ion modes were adopted for ion acquisition.

MS2 was performed immediately after the LC-MS1 experiment with the corresponding mode (C18-ESI⁺, C18-ESI⁻, HILIC-ESI⁺, or HILIC-ESI⁻). The chromatographic conditions were the same as the ones detailed in the LC-MS section, while the mass spectrometer was specifically configured for untargeted DDA, DIA, and iterative targeted DDA, respectively. DIA was performed with sequential window acquisition of all theoretical fragment ion mass spectra (SWATH-MS) strategy. All parameters for the MS2 acquisitions are summarized in Supplementary Table 10.

Untargeted DDA was performed to detect the top 10 ions with the highest intensities of each full MS scan. Immediately after the acquisition of MS1 and untargeted DDA, SWATH was performed. Each cycle of SWATH consisted of a full MS scan and 10 MS2 windows with different window sizes. The *m/z* size of the MS2 window was determined based on the general distribution of metabolic features at the MS1 level. The adjacent windows were sequentially used for DIA MS2 detection. The windows overlapped with their neighbors at 1.0 *m/z*. The design of SWATH windows is summarized in

Supplementary Table 11. Approximately 0.9 s elapsed in total for each SWATH cycle.

The sample type-specific metabolic features were extracted based on the results of MS1 and used as the inclusion list. Iterative targeted DDA was executed with the updated inclusion list (optimized using HERMES[60]) to exhaust the targets as much as possible[61]. In detail, the sample-specific ions were input as the acquisition targets for DDA. Once the 1st round of DDA was finished, the detected ions were excluded from the target inclusion list. Then, the 2nd round of targeted DDA was performed with the updated inclusion list until the targets or samples were exhausted. All raw spectra data were deposited to the Metabolomics Workbench (accession ID: ST002796 and ST002798).

**MS2 reference spectra library curation**

A total of nine public MS2 databases were collected and curated. HMDB databases were downloaded directly from its website as xml files. Four tables (HMDB_experimental_NegDB, HMDB_experimental_PosDB, HMDB_predicted_NegDB, HMDB_predicted_NegDB) were generated from the HMDB database. MoNA series and

LipidBlast database were downloaded from MassBank of North America as msp files. Nine tables (MoNA_PosDB, MoNA_NegDB, ReSpect_PosDB, ReSpect_NegDB, VaniyaNP_PosDB, Vaniya_NegDB, BMDMS_PosDB, LipidBlast_PosDB and LipidBlast_NegDB) were generated. MassBank database was downloaded from its website as msp files. Four tables were generated (RIKEN_PosDB, RIKEN_NegDB, MassBank_PosDB, MassBank_NegDB). GNPS database was downloaded from the GNPS website as msp files. Two tables (GNPS_PosDB, GNPS_NegDB) were generated. MINEs database was downloaded from the MINEs website in msp format. Two tables were curated (MINEs_PosDB, MINEs_NegDB) from it. These downloaded files were curated with in-house R scripts into SQLite format based on the same schema, which is named "Complete library". Detailed information regarding downloading links, database versions, and access dates are available in Supplementary Table 12. Based on the "Complete library", we derived four specific MS2 reference libraries (Pathway library, Biology library, Lipid library, and Exposomics library) as described below.

The pathway library was mainly curated according to the KEGG pathway information. KEGG pathways of 120 species (including common model species and pathogenic microorganisms) were downloaded with KEGGREST[62]. All compounds from the metabolic pathways of all species were extracted as a compound list. A total of 3456 compounds were included (supplementary Fig. 28). All MS2 records in the "Complete library" matching these compounds were extracted as a "Pathway library". Similarly, Biology library was curated based on the compound information from KEGG[17] compound database and HMDB[31]. All compounds and glycans from KEGG are summarized as compound list 1. All compounds in HMDB labeled as "Serum", "Urine", "Sweat", "Saliva", "Feces" and "Cerebrospinal Fluid" were summarized as compound list 2. Compound lists 1 and 2 were merged as a target list. All MS2 records in the "Complete library" matching these compounds were extracted as a "Biology library". The lipid library was curated based on the compound information from LIPIDMAPS, LipidBank[34], LipidBlast[33], and HMDB database[63]. All lipids from these databases were summarized as a lipid list. All MS2 records in the "Complete library" matching these compounds were extracted as a "Lipid library". All lipids in this library are classified into superclasses, main classes, and sub-classes based on RefMet[64]. Exposomics library was curated from KEGG Drug database[65], Microbial Metabolites Database[66], Toxin-Toxin-Target Database (T3DB)[67], FooDB (www.foodb.ca), Phenol-Explorer[68], Exposome-Explorer[69], and NORMAN Suspect List Exchange database[70]. All compounds from these databases were extracted as exposomics compound list. All MS2 records in the "Complete library" matching these compounds were separately extracted as an "Exposomics library".

Five neutral loss spectra databases have also been pre-calculated, corresponding to the five options above. The curation of neutral loss databases was based on the algorithm implemented by METLIN neutral loss database[30]. Briefly, neutral loss spectra were calculated by deducting the $m/z$ from the precursor ion as the neutral loss ion. The intensity values were directly mirrored. The scheme of all MS2 reference libraries is shown in Supplementary Fig. 29.

## DDA data deconvolution

The first step of DDA spectra deconvolution is to assign all MS2 spectra to individual target features based on the precursor information. If users provide a target feature list, MetaboAnalystR can automatically perform MS2 data processing. Otherwise, the complete MS1 features will be used for MS2 data processing. By default, the MS1 features list generated by MetaboAnalystR includes minimum and maximum values for $m/z$ and retention time. If $m/z$ and retention time are not provided as in ranges, the $m/z$ and retention time ranges will be calculated automatically based on

tolerance values defined by users (ppm for $m/z$, and rt_tol for retention time). If there are multiple MS2 spectra assigned to an individual target MS1 feature, the spectra would be merged in a weighted manner as developed in MZmine[9]. The median $m/z$ (mz_med) and median retention time (rt_med) are extracted or calculated based on the MS feature's information. Then the nearest MS1 is extracted for the following analysis. If there are multiple different centroids within the (mz_med centered) isolation window, and any of their intensity values are over the acquisition threshold (user-defined), the spectrum is considered as "chimeric" to be deconvolved in the next steps. The centroid ion corresponding to the MS feature is considered as "main ion", others are considered "contamination ions".

The purpose of deconvolution is to remove the fragments produced from "contamination ions" in the chimeric spectrum ("Spectrum 0") and generate a clean deconvoluted spectrum for the "main ion". Technically, the "main ion" is the ion of target feature, while the "contamination ion" may come from multiple sources, such as isobaric ions, orphan isotopologues[18], other known or unknown ions with their $m/z$ values falling into the isolation windows, etc. In the second step, MetaboAnalystR scans through all contamination ions and determines if they are orphan isotopologues. If any of them is identified as orphan isotopologues based on the MS1 scan, the spectrum of this orphan isotopologue is predicted with the method from DecoID[18]. We name the predicted spectrum of the orphan isotopologue as "Candidate Spectrum I". Then, if there is any ion that has been detected and identified as a clean spectrum in one or two nearest MS2 scans inside the data itself, the clean spectrum is also extracted for deconvolution, named "Candidate Spectrum II". Next, MetaboAnalystR extracts potential spectra from the MS2 library as the reference spectrum for the ions, which are neither orphan isotopologues nor the ones with clean spectra included by the spectra data itself. All spectra from the reference are extracted, and the one showing the highest similarity to the original chimeric spectrum ("Spectrum 0") is retained as "Candidate Spectrum III". The spectral similarity is evaluated with dot product[18] or spectral entropy[29] similarity methods based on users' preferences. If all ions in this isolation window have been assigned with a reference spectrum, the deconvolution can be executed directly.

In many cases, however, some ions can be identified as neither orphan isotopologues nor the ones with a clean spectrum contained in the data itself nor the ones with a reference spectrum from the library. These ions will be named "Unknown ions". MetaboAnalystR predicts the spectrum of these "Unknown ions" based on the hypothesis that ions with abiotic/bio-transformation relationships will share highly similar MS2 spectra pattern[2,28]. In this case, the most accurate formula is first predicted for the "Unknown ion". Then an abiotic/bio-transformation network is constructed around the "Unknown ion" (Fig. 1c) based on the rules from NetID[28]. Different from the network used in NetID, this prediction network model is not propagatable to avoid potential redundancy. Once the network is constructed, all neighbors of the "Unknown ion" are searched against the library to get their spectra data. All fragments of the spectra data are predicted as the most accurate formula. If the chemical elements' composition of the formula is against the formula of the "Unknown ion", this fragment is considered as an unreasonable fragment and will be removed from the spectrum. Once this cleaning step is completed, the similarity of all spectra to the original chimeric spectrum ("Spectrum 0") is evaluated. The one with the highest similarity score is returned as the predicted spectrum for the "Unknown ion". It is named "Candidate Spectrum IV".

Next, Candidate Spectra I–IV are returned as the components to deconvolve the original chimeric spectrum ("Spectrum 0"). Given that Candidate, Spectrum IV is neither from real data nor a reference

library, a penalty (0–10; 0, no penalty for the perfect match; 10, 10 times penalty for the negative match) is given based on the similarity to the "Spectrum 0". Deconvolution on the "Spectrum 0" is performed with a penalized elastic-net regression model[71,72]. The purpose of this deconvolution model is to minimize the residue. The deconvolution method is shown as the formula below:

$$Residue = \min\left(\sum_{i=0}^{n}(y - x_i\beta)^2 + \lambda P_\alpha(\beta)\right) \tag{1}$$

where $P_\alpha(\beta) = \frac{1}{2}(1-\alpha)||\beta||_2^2 + \alpha||\beta||_1$ is the elastic-net penalty[72]. y is the response vector (Spectrum 0), x is the candidate components (Candidate Spectra I–IV). In this model, $\alpha$ and $\lambda$ are two critical parameters. If $\alpha = 1$, the model is a LASSO regression model, similar to DecoID[18]. Instead of using an arbitrary value for $\alpha$ and $\lambda$ from DecoID, MetaboAnalystR permutates a matrix of $\alpha$ and $\lambda$ combination. In detail, 11 $\alpha$ values (starting from 0, and end with 1, step by 0.1) and 10 $\lambda$ values, estimated by the correlations between response vector and component vectors[72]. Therefore, 110 $\alpha$ and $\lambda$ combinations are prepared to optimize the elastic-net model in an automated manner. Then, deconvolution based on the penalized elastic model is executed and results in 110 solutions for "Spectrum 0". All residues of 110 solutions are iterated and the one with minimal residue is returned (Solution 0). Different from DecoID, MetaboAnalystR optimizes $\alpha$ and $\lambda$ for every individual peak, instead of implying a hard value for all peaks.

Finally, all $\beta$ values for contamination ions ($\beta_{contms}$) in Solution 0 is used to remove fragments in "Spectrum 0". The remaining fragments are normalized and exported as "deconvoluted" spectrum for "main ion". If there is no fragment left after the cleaning or the $\beta$ value for "main ion" is 0, the deconvolution failed. The original "Spectrum 0" will be retained and exported directly for MS2 reference library searching. The deconvolution of DDA data can be achieved with the function, *PerformDDADeconvolution*.

## SWATH-DIA data deconvolution

SWATH-DIA data deconvolution algorithm follows the steps described by DecoMetDIA[23]. In brief, for a specific MS feature, all extracted ion chromatograms (EICs) of MS2 peaks from the corresponding SWATH window are detected and clustered based on peak similarity and retention time information. One model peak is selected from each cluster, and all model peaks are organized to decompose all EICs. Each decomposed component from different EICs is used to reconstruct the composition of the MS2 cluster. The cluster containing the original MS feature is exported as a *pseudo*-MS2 spectrum. Unlike DecoMetDIA, the entire data deconvolution workflow is implemented in Rcpp/C++. The deconvolution of SWATH-DIA data can be achieved with the function, *PerformDIADeconvolution*.

## Spectra consensus for replicates

MS2 data are commonly acquired with multiple replicates. In such cases, all deconvolved MS2 spectra corresponding to the same MS1 peak must be processed to generate a single consensus spectrum. If there are no replicates, this step is skipped. All MS2 fragments across different replicates are initially summarized by count. If the frequency of an individual fragment is above a user-defined threshold (e.g., 50%), it is kept; otherwise, the fragment is removed. Optionally, a database-assisted spectrum consensus can be used to assist the process (see Supplementary Fig. 1). If database-assisted option is enabled by users, all spectra of the precursor are extracted as referent list (L) from the reference library. All fragments not meeting the user-defined frequency threshold is then searched against L. If this fragment can be found from L and the frequency of the fragment across the replicates is over 2, the fragment is kept; otherwise, it is discarded. All the remaining fragments are normalized and merged to generate a consensus spectrum for database searching in the next step. The spectrum consensus can be achieved with the function, *PerformSpectrumConsensus*.

## Reference library searching and scoring

Reference library searching is based on the *m/z* (and optionally, the information on retention time) of precursors. All matches are extracted from the database for scoring. MetaboAnalystR uses the same scoring rule as MS-DIAL[5]. The matching score is calculated using the following formula:

$$\begin{aligned} &Matching\ Score \\ &= \frac{MS2\ Similarity + MS1\ Similarity + RT\ Similarity + 0.5 \times Isotope\ similarity}{3.5} \times 100 \end{aligned} \tag{2}$$

where the MS2 similarity (ranging from 0 to 1) is calculated using the popular dot product similarity[5,18] or spectral entropy similarity[29] algorithms. MS1 similarity and retention time (RT) similarity (both ranging from 0 to 1) are calculated with an exponential distribution method based on deviation. Isotope similarity is also calculated using a similar method as implemented in MS-DIAL. However, the calculation of isotope distribution similarity only considers $[M+n]$ $(n<3)$, as the intensity of isotopes $[M+n]$ $(n\geq3)$ is very low and highly variant. Briefly, the similarity evaluation of isotope distributions is performed based on the experimental isotope distribution and the theoretical distribution of formulas extracted from the reference library. Isotope elements considered here include carbon $(C^{13})$, hydrogen $(H^2)$, nitrogen $(N^{15})$, oxygen $(O^{17}, O^{18})$, and sulfur $(S^{33}, S^{34})$. Other elements are not considered due to their extremely low abundance in nature. Retention time is optionally used based on users' requests. If retention time is disabled (by default), the retention time similarity is not calculated, and the denominator is modified to 2.5. The database searching is performed via SQLite query and can be achieved with the function, *PerformDBSearchingBatch*.

## Neutral loss searching

Neutral loss presents a mirror of MS2 spectra data. They are discarded fragments without charges during ionization and detection. Neutral loss is valuable for characterizing unknown compounds[30]. If the matching score is below 10 (out of a maximum of 100), the neutral loss searching option can be performed to find the potential chemical identification. In such cases, the target precursor (P0) is extracted and incorporated into the abiotic/bio-transformation network model[28] (described in "DDA data deconvolution algorithm") to find all potential neighbors. These neighbors are used as targets for extracting spectra from the neutral loss reference library. The neutral loss of P0 is calculated directly[30] and is matched against the neutral loss spectra extracted from the reference library. All potential matches are scored and exported as a reference of chemical identification. The results are labeled as "Neutral loss matching". The database searching with neutral loss can be achieved by enabling the "enableNL" parameter in the *PerformDBSearchingBatch* function.

## Result export

All compound identification results can be exported as a data frame in the form of a .csv or .txt file. The exported information includes compound names, chemical formula, InChIKeys, and matching scores. If the reference library is a lipid library, the exported information also includes lipid classifications (superclass, main class, and sub-class). When multiple matches are returned for a specific feature, MetaboAnalystR will check for those that are chemically the same (i.e. due to redundancies in database records, see Table 1) and only keep the one with the highest matching score. The database search results can be exported as a data frame table using the *PerformResultsExport* function.

## Decoy spectra generation and null evaluation

To generate decoy spectra, a mixture of 91 compounds[40] was used as the raw spectra data in the mzML format, which was imported using the mzR package[73]. Spectral scans were split into MS1 data and MS2 data based on MS level (as shown in Fig. 3c). MS data was processed similarly for both DDA and SWATH-DIA methods. Specifically, the $m/z$ values of mass centroids of MS data were randomly adjusted by adding mass errors ranging from 10 to 30 ppm, while the intensity values were randomly distorted by multiplying with a coefficient (ranging from 0.01 to 50.0). The retention time dimension was kept unchanged. For DDA spectra, the MS2 spectrum pattern was replaced with a synthetic MS2 spectrum randomly simulated from isobaric compounds. For SWATH-DIA spectra, the MS2 spectrum pattern was processed in the same way as MS spectra, while the SWATH window and retention time were kept unchanged. A total of 18 spectra decoy spectra data were generated for each replicate in both DDA and SWATH-DIA datasets. These decoy spectra were processed in the same way as the original real dataset using different tools to perform *null* evaluations. After processing, all compounds within the decoy samples are categorized as "true negatives". If they are still identified as their original identities, they are classified as "false positives". Evaluations based on these decoy samples were conducted for all tools with their database options, encompassing all original compounds.

## MetaboAnalystR usage

The MS spectral data used in this study were converted into mzML centroid mode using Proteowizard[73] for both MS1 and MS2 levels. The auto-optimized workflow was first applied to process the MS spectra, including peak picking, peak alignment, gap filling, and peak annotation, to generate complete MS feature tables. These tables were used as the target list for MS2 spectra processing in both DDA and SWATH-DIA. The chemical classification analysis of blood samples was performed using ClassyFire[63]. The R stats package was employed to perform Pearson correlation analysis of serial dilutions. The heatmap analysis was directly performed using MetaboAnalystR. Identification and matching of compounds to the list of standards were based on the exact matching of InChIKeys, deviation of retention time less than 20 s with mass tolerance 5 ppm.

## MS-DIAL/MS-FINDER usage

MS-DIAL (v4.9.22, Linux version) and MS-FINDER (v3.52, Linux version) were used. A mass accuracy parameter of 0.005 Da was set for "MS1 tolerance" and 0.01 Da for "MS2 tolerance". The minimum percentage of peaks within one group was set to 50%, while other parameters were left as default. Following DDA or SWATH-DIA data processing, peak area alignment results were exported as the results of the MS level. All features with MS2 information were exported for MS-FINDER analysis performed in batch mode. PubChem remote query was only allowed when there was no candidate from other databases. All MS2 spectra libraries were selected for searching, while other parameters were left as default values. The identified formula and structures were exported automatically by MS-FINDER. Compound identification and matching to the list of standards were based on InChIKeys.

## MZmine usage

We used MZmine (v3.2.8, Linux version) to process the raw spectra data. Firstly, we imported raw data and performed mass detection at the MS1 level. Next, we executed the ADAP chromatogram builder[74] with a parameter scan-to-scan accuracy set at 0.005 Da or 10 ppm, while keeping other parameters as default values. We then performed smoothing and joint alignment to generate an aligned feature list and exported as the MS1 feature table. Subsequently, we executed the MSn feature list builder, followed by mass detection at the MS2 level.

Finally, the MSn feature list was exported to the SIRIUS/CSI:FingerID format with all MS2 feature lists selected. We enabled the merge MS2 option, leaving the other parameters as default values in this step. To evaluate computational performance, we processed the COVID-19 dataset in batch mode from the command line. For other datasets, we used the MZmine UI for analysis.

## XCMS usage

XCMS (v3.20.0, an R package) was used to process SWATH-DIA data. All parameters from the XCMS online platform (https://xcmsonline.scripps.edu/) were used to process MS1 data. For SWATH-DIA data processing, we utilized the function, *reconstructChromPeakSpectra*, to deconvolve SWATH-DIA data. We exported all deconvoluted spectra into a.msp file using an in-house R script.

## SIRIUS usage

We used SIRIUS (v5.6.3, Linux headless version) for the MS2 spectra search. For formula prediction, we set the program to use the entire database, while enabling ZODIAC, CSI:FingerID, and CANOPUS. For CSI:FingerID, we selected all available databases. Other parameters were left as their default values. We exported the MS2 searching results with the "write summarize" option enabled. To identify chemicals and match them to the standards list, we used the InChIKeys generated by InChIs from SIRIUS.

## Assessment of computational performance

We used SLURM (v22.05.6) to execute and record the usage of the computational resources for each job. All tools had a command-line interface to be executed. We allocated two CPU cores and all RAM resources for each tool for comparison. We recorded the clock time between the starting and ending of the job, as well as the maximum usage of RAM.

## Integration of MS2 results into the *mummichog* algorithm

MetaboAnalystR can process LC-MS1 and MS2 spectra directly and format the results for pathway enrichment prediction with *mummichog*. It also accepts MS1 peaks list/table individually or in combination with MS2-based compound identification. The *mummichog* algorithm was improved by incorporating MS2-based chemical identification results. Initially, the algorithm matches all features based on their $m/z$ and/or retention time to generate empirical compounds[27]. One MS1 feature may be mapped to multiple empirical compounds, and MS2-based chemical identifications will be utilized to filter out those unfeasible candidates to produce a shorter but more accurate list of empirical compounds. The permutation test is then performed based on the filtered list. The underlying pathway libraries have been updated with additional compound IDs to be more compatible with the results generated from MS2 identification. In MetaboAnalystR 4.0, the compounds in pathway databases have been converted into different types of chemical IDs, including InChIKeys, KEGG IDs, HMDB IDs, PubChem SIDs, PubChem CIDs, and SMILES.

## Interfacing with other tools and databases

MetaboAnalystR can also accept results from other well-established raw spectra processing tools, such as MS-DIAL (in mat format), MZmine (in msp format), and XCMS (as an object in R) for MS2 identification with a comprehensive/specific database. In addition, MetaboAnalystR can automatically convert MS2 identification results from MS-FINDER (structure result table) and SIRIUS (compound identification table) into a compound list for *mummichog*-based pathway enrichment analysis. All MS2 reference libraries are curated as SQLite libraries. Users can easily convert their in-house reference library into SQLite format to be included in the MetaboAnalystR workflow.

## Reporting summary

Further information on research design is available in the Nature Portfolio Reporting Summary linked to this article.

## Data availability

LC-MS and MS2 data of blood samples and serial dilutions used in the evaluation of MetaboAnalystR 4.0 have been uploaded to Metabolomics Workbench (https://www.metabolomicsworkbench.org/) as studies ST002796 and ST002798. The complex standards mixture data was obtained from MetaboLights repository (ID: MTBLS2207). The simple standards mixture series data was from Curatr (https://curatr.mcf.embl.de/). The standard mixtures with medium complexity were also from MetaboLights repository (ID: MTBLS1311). Polar and non-polar DDA metabolomics datasets of COVID-19 were downloaded from MetaboLights repository (ID: MTBLS2542). The COVID-19 SWATH-DIA metabolomics dataset was obtained from MassIVE repository (ID: MSV000089568 [https://doi.org/10.25345/C5HQ3S30D]). All curated MS2 spectral libraries and descriptions are available from the MetaboAnalyst website (https://www.metaboanalyst.ca/docs/Databases.xhtml). Source data are provided with this paper.

## Code availability

Source code of MetaboAnalystR 4.0 is available from GitHub (https://github.com/xia-lab/MetaboAnalystR). The codes used for benchmark studies and analysis is available also from GitHub (https://github.com/zhiqiang-PANG/MetabR4_scripts). A step-by-step tutorial of MetaboAnalystR 4.0 has been provided in the MetaboAnalyst website (https://www.metaboanalyst.ca/docs/RTutorial.xhtml).

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

## Acknowledgements

This project was jointly funded by multiple grants (to J.X.), including Genome Canada, Genome Quebec, US National Institutes of Health (U01 CA235493), Natural Sciences and Engineering Research Council of Canada (NSERC), Canada Research Chairs (CRC) program, and Canada Foundation for Innovation (CFI).

## Author contributions

This project was conceived and designed by Z.P., and J.X; The novel algorithms and methods were developed by Z.P. and J.X; Formal analysis and data curation was performed by Z.P., L.X., C.V., Y.L., N.B., and J.X.; Experiment and sample preparation was performed by C.V. and Z.P.; Software was validated and evaluated by Z.P. and L.X.; N.B. contributed experimental resources for validation; The original draft was written by Z.P., N.B., and J.X., and all authors contributed to reviewing and editing of the final manuscript. J.X. acquired funding and supervised the project. R.S. provided draft editing and supervision on the project.

## Competing interests

J.X. is the founder of XiaLab Analytics. The remaining authors declare no competing interests.
