## [Peer Review File · Nature Communications]

Reviewers' Comments:

Reviewer #1:

Remarks to the Author:

The authors present an exciting update to the popular MetaboAnalyst open-source tool to support metabolomics data analysis. The MetaboAnalyst platform is widely adopted by the community as evidenced by the 1000+ citations to date. The update reported in the current manuscript is in response to user requests for end-to-end data processing. The authors present sufficient benchmarking results to support that the platform performs comparably to the two tools chosen for evaluation. However, the gains are incremental relative to unoptimized performance of other tools. The challenge of metabolomics data analysis is that we don't know what the right answer is – and there is not one data analysis tool that performs best for all data types and sources. Rather than focus on a narrative of “my tool is better than yours”, I would like to see the manuscript focus on what sets MetaboAnalyst 4.0 apart from other tools, namely the end-to-end processing from raw data through stats and pathway analysis and faster, more efficient computation in the R environment. In addition the flexibility of this new iteration of MetaboAnalystR to integrate data processed using other tools can be imported at various points in the workflow should be elevated a -it would be great to have more discussion and a figure describing at which points in the workflow this can happen. Finally, the lack of interface to visualize spectral matches is a huge limitation and requires that users blindly accept annotation results without any way to confirm or validate the spectral match or have any insight into the confidence level of annotations. I would like to see this limitation resolved prior to publication.

Specific comments:

- The manuscript should be reviewed thoroughly for English grammar. There are numerous instances of missing articles (the, a, an) and spelling mistakes which are distracting for the reader. Additionally, m/z should be italicized in text and figures.
- Abstract: consider reframing the value of MetaboAnalstR 4.0 to address the need for an end-to-end tool for metabolomics data analysis. Remove specifics from benchmarking – these are misleading as other tools were evaluated using only default parameters (and the evaluated tools are not comprehensive of ALL other common metabolomics tools). Suggest that this be condensed to a more general statement such as: “Benchmarking against other well-established platforms demonstrated that MetaboAnalstR 4.0 performs as well or better for tasks such as peak picking, spectral deconvolution and compound annotation.”
- Introduction: this section needs to be completely reorganized. What is the key need you are addressing? It is not DDA deconvolution – yet that is the first paragraph of the introduction. There is also a fragmented and non-comprehensive review of existing tools for DIA and DDA deconvolution – this section is not needed. Again – the primary gap you are filling is an end-to-end solution.
- Line 106: do you mean “.....all MS2 spectra from a single sample into different.....”?
- Line 110: it is unclear what is referred to as “reference libraries”. Is this referring to the compiled spectral libraries? What if there are multiple matches to the reference library? This step is very unclear since it is prior to the database searching step for compound annotation.
- Line 118: do you mean “.....all MS2 spectra from replicates.....”?
- Line 120: What do you mean by “This process can optionally be assisted by the observed frequencies of individual fragments in a spectral database”?
- Lines 131-141: this section is very important and should be emphasized more. The compiled reference spectral databases and the ability to prepare and use custom libraries is another unique feature that is undersold in the manuscript.
- Line 166-168 change “1st SM data” to “1st SM dataset” (same for 2nd and 3rd) – use this throughout the text.
- Line 168: do you mean that the 2nd SM dataset was analyzed by both DDA and SWATH-DIA (ESI+) in separate injections (single replicate, ESI+ only)?
- Line 168-170: rephrase for clarity “The 3rd SM dataset contains 91 compounds (medium complexity) and was analyzed by both DDA and SWATH-DIA (three replicates each in ESI+ and ESI-).”
- Line 172: delete “always”, change “most correct” to “the correct”. What is the compound discovery ratio? How is this just over 0.75 in Figure 2a but you are saying that the correct compound was the top candidate 100% of the time?
- Line 176: highest number of MS features does not mean correct features.

- Line 177: what does "efficient way" mean?
- Lines 189-197: suggest removing this or moving to supplemental. It is not relevant.
- Figure 2c: These are not valid comparisons since you are just using the default settings. I think it is more important to show how your tool performs (i.e., annotation accuracy with the known SM datasets) not how it performs arbitrarily to other tools. Additionally, the libraries used across tools is ambiguous to the reader and could be a biasing affect to these results. Suggest removing lines 208-215.
- Section starting at Line 217 and including Figure 4. Suggest focusing this on performance of MetaboAnalystR 4.0 and not on the comparison with other tools.
- Likewise with the section starting on line 243. Suggest that these two sections could be combined and condensed significantly. Figures 4-5 could be moved to supplemental.
- Line 263: it is not important that these are "COVID-19" datasets. Suggest generalizing this section by reframing as "Biological interpretation of clinical metabolomics data".
- Line 266: this should be the highlighted point of the manuscript! It is buried here in this section.
- Line 276: highlight the automated parameter tuning without comparison to other tools. You can just highlight this as an advantage of your tool.
- In general, the # of features detected is not diagnostic of "correctness" – especially in a complex sample where we do not know what "correct" is. This is also true for pathway prediction (line 291) – just predicting "more" pathways does not mean better.
- Line 301: change "COVID-19" to "clinical metabolomics" (or similar).
- I do not feel that the conclusions in the discussion are supported by what is presented. For example, line 331 – you cannot claim that you have more "accurate functional enrichment analysis". Again, it would be more appropriate to focus on the capabilities of your tool not the comparison to other tools.

Reviewer #2:

Remarks to the Author:

Reviewer #3:

Remarks to the Author:

This manuscript is a comprehensive report on the performance of the MetaboAnalystR 4. The amount of conducted studies is overwhelming and is also impressive. The authors should have spent much time and effort to produce this much results and assessments. I really admire this thorough effort to improve the efficiency of metabolomics research.

This said, my overall impression of this manuscript is that the question on novelty: what is the strong benefit or the advancement in terms of science? Maybe the power of integration is one answer, but the current manuscript fails to address the true novelty of the contribution because the performance is rather competing than to be called excelling.

The novelty in my understanding is the peak-wise introduction of the elastic-net optimization, assessing as many as 110 parameter combinations for every detected peak (Page 26). Honestly, I cannot believe that this much computation can compete with existing software platforms in time, but anyway, this is what is described. So if the authors wish to elaborate on the scientific novelty, this optimization part should be the focal point. My doubt is that the elastic-net may be unnecessary; the performance of identification may be equivalent by just using LASSO. My suggestion for the authors is that they should focus more on such methodological improvement, rather than a superficial competition between software platforms. In my opinion, comparison with MS-DIAL/MS-FINDER or MZmine/SIRIUS is futile because the software performance hugely depends on parameter optimization (of course!), and the default setting is not good at all. In other words, we should NOT compete in performance but in rigorous methodology / theory. Since the current manuscript leans too much toward beating other software platforms, the manuscript became unnecessarily lengthy and descriptive on actual performance. My preference is to bring the Method part more to the front-end and push aside the actual performances in

supplements.

Major recommendation:

1. Rather than focusing on the relative performance of existing software platforms, the authors need to focus on the theoretical / methodological advantage of this contribution.
2. This means that description on the neutral-loss searching and the replicate treatment (Page 27-28) needs elaboration, because the current explanation is hard to understand (e.g. what is the NL reference library and how multiple matches are handled).
3. The decoy library is not really a decoy, since it's a standard mixture. A more detailed explanation is needed to accurately evaluate the decoy performance, since this work includes not only small metabolites but lipids and exposome metabolites.
4. Availability of blood sample data and software R code is welcome but researchers also need data. Is it possible to share curated data?
5. Description of the mummichog algorithm is missing. Please describe at least its overview in the Method section (P32 is not enough at all).
6. Please add availability of curated spectral data.

Reviewer #1 (Remarks to the Author):

The authors present an exciting update to the popular MetaboAnalyst open-source tool to support metabolomics data analysis. The MetaboAnalyst platform is widely adopted by the community as evidenced by the 1000+ citations to date. The update reported in the current manuscript is in response to user requests for end-to-end data processing. The authors present sufficient benchmarking results to support that the platform performs comparably to the two tools chosen for evaluation. However, the gains are incremental relative to unoptimized performance of other tools. The challenge of metabolomics data analysis is that we don't know what the right answer is – and there is not one data analysis tool that performs best for all data types and sources. Rather than focus on a narrative of “my tool is better than yours”, I would like to see the manuscript focus on what sets MetaboAnalyst 4.0 apart from other tools, namely the end-to-end processing from raw data through stats and pathway analysis and faster, more efficient computation in the R environment. In addition the flexibility of this new iteration of MetaboAnalystR to integrate data processed using other tools can be imported at various points in the workflow should be elevated a -it would be great to have more discussion and a figure describing at which points in the workflow this can happen. Finally, the lack of interface to visualize spectral matches is a huge limitation and requires that users blindly accept annotation results without any way to confirm or validate the spectral match or have any insight into the confidence level of annotations. I would like to see this limitation resolved prior to publication.

Response: Thank you for your encouraging comments and suggestions on improving our manuscript as well as MetaboAnalystR.

- We have systematically rewritten the entire manuscript based on your advice.
- We have minimized the comparison to other tools. Most comparison content has been removed or moved to supplements. If any comparison is necessary, we have clearly stated that other tools are performed with their default settings.
- We have implemented an interactive mirror plot for users to explore and compare their compound identification results. An example has been provided in **Figure 4e**.

Specific comments:

- The manuscript should be reviewed thoroughly for English grammar. There are numerous instances of missing articles (the, a, an) and spelling mistakes which are distracting for the reader. Additionally, m/z should be italicized in text and figures.

Response: We have thoroughly checked and corrected all English grammar of this manuscript. All m/z symbols have also been italicized based on the standard.

- Abstract: consider reframing the value of MetaboAnalstR 4.0 to address the need for an end-to-end tool for metabolomics data analysis. Remove specifics from benchmarking – these are misleading as other tools were evaluated using only default parameters (and the evaluated tools are not comprehensive of ALL other common metabolomics tools). Suggest that this be condensed to a more general statement such as: “Benchmarking against other well-established platforms demonstrated that MetaboAnalstR 4.0 performs as well or better for tasks such as peak picking, spectral deconvolution and compound annotation.”

Response: Thanks for this advice. We have re-organized the abstract and removed all benchmarking details and relocated them to the supplementary material. We added the condensed sentence instead.

- Introduction: this section needs to be completely reorganized. What is the key need you are addressing? It is not DDA deconvolution – yet that is the first paragraph of the introduction. There is also a fragmented and non-comprehensive review of existing tools for DIA and DDA deconvolution – this section is not needed. Again – the primary gap you are filling is an end-to-end solution.

Response: Yes, we have carefully rewritten the introduction section according to your suggestions. In this revised introduction, we have emphasized that the highlight of this study, MetaboAnalystR 4.0, is its unified workflow, rather than dwelling extensively on DDA or DIA. The content related to DDA/DIA has been significantly reduced as one of the key issues addressed in this version.

- Line 106: do you mean “.....all MS2 spectra from a single sample into different.....”?

Response: Sorry for the confusion. Your understanding is correct. We have rephrased the sentence based on this advice.

- Line 110: it is unclear what is referred to as “reference libraries”. Is this referring to the compiled spectral libraries? What if there are multiple matches to the reference library? This step is very unclear since it is prior to the database searching step for compound annotation.

Response: Yes, the reference libraries refer to the MS2 spectral database. We have revised the sentence to clarify this point in the manuscript. If there are multiple matches returned, the one with highest similarity to the original spectrum will be kept. To avoid bring up too many methodological details at the results section, we have provided the corresponding description on this point in the Method section (as below).

‘All spectra from reference are extracted, and the one showing highest similarity to the original chimeric spectrum (“Spectrum 0”) is retained as “Candidate Spectrum III”’

- Line 118: do you mean “....all MS2 spectra from replicates....”?

Response: Yes, it refers to MS2 spectra. We have corrected this.

- Line 120: What do you mean by “This process can optionally be assisted by the observed frequencies of individual fragments in a spectral database”?

Response: Thank you for the question. We have rephrased the sentence and provided more necessary details for this step to make it clear.

- Lines 131-141: this section is very important and should be emphasized more. The compiled reference spectral databases and the ability to prepare and use custom libraries is another unique feature that is undersold in the manuscript.

Response: Thanks for this comment. We have rewritten this part. First, we have provided more information on the compatibility with msp files. As for the database, we fully agree with you. The

paragraph has been separated as an independent subsection. More details have also been added for both available MS2 reference database and customizable option for users.

- Line 166-168 change “1st SM data” to “1st SM dataset” (same for 2nd and 3rd) – use this throughout the text.

Response: We have corrected this throughout the whole manuscript. For both 1st, 2nd, 3rd and following clinical (COVID-19) datasets.

- Line 168: do you mean that the 2nd SM dataset was analyzed by both DDA and SWATH-DIA (ESI+) in separate injections (single replicate, ESI+ only)?

Response: The 2nd SM was analyzed by both DDA and SWATH-DIA in separate injections with only one replicate. But they are analyzed in both ESI⁺ and ESI⁻. We have corrected the corresponding description of the manuscript. Please note that we have re-organized all benchmark and case studies in this revision. The order for 1st and 2nd SM dataset has been exchanged and renamed.

- Line 168-170: rephrase for clarity “The 3rd SM dataset contains 91 compounds (medium complexity) and was analyzed by both DDA and SWATH-DIA (three replicates each in ESI+ and ESI-).”

Response: Thanks for pointing out this. We sentence has been rephrased as you suggested.

- Line 172: delete “always”, change “most correct” to “the correct”. What is the compound discovery ratio? How is this just over 0.75 in Figure 2a but you are saying that the correct compound was the top candidate 100% of the time?

Response: We have deleted these confusing descriptions from the manuscript completely. The evaluation on MS2 data processing performance has been re-organized. The results have been simplified tremendously to avoid such confusion.

- Line 176: highest number of MS features does not mean correct features.

Response: We have deleted this description from the manuscript.

- Line 177: what does “efficient way” mean?

Response: “efficient way” means highly computational efficiency. We have rephrased the sentence to avoid such confusion.

- Lines 189-197: suggest removing this or moving to supplemental. It is not relevant.

Response: Yes. We have completely removed this subsection. The effects of reference database are not the key point of this manuscript.

- Figure 2c: These are not valid comparisons since you are just using the default settings. I think it is more important to show how your tool performs (i.e., annotation accuracy with the known SM

datasets) not how it performs arbitrarily to other tools. Additionally, the libraries used across tools is ambiguous to the reader and could be a biasing affect to these results. Suggest removing lines 208-215.

Response: Thank you for the comments. We have also realized this point. The manuscript has been rewritten to focus on how MetaboAnalystR behaves and supports outputs from other spectra processing tools. We have minimized comparisons to other tools, mentioning them sparingly and consistently noting that they are being evaluated under their default settings.

- Section starting at Line 217 and including Figure 4. Suggest focusing this on performance of MetaboAnalystR 4.0 and not on the comparison with other tools.

Response: Yes. This section has been removed. The blood sample dataset has been used to demonstrate the performance of LC-MS processing in MetaboAnalystR. All unnecessary comparisons have been deleted directly or moved to supplementary.

- Likewise with the section starting on line 243. Suggest that these two sections could be combined and condensed significantly. Figures 4-5 could be moved to supplemental.

Response: Agreed. We have combined the evaluation with blood samples and serial dilutions as a part of evaluation of LC-MS1 data processing performance. The unnecessary content and figures have been condensed significantly based on your advice. Key figures from original Figure 4 and 5 have been re-organized into Figure 3. Others have been moved to supplementary materials based on your advice.

- Line 263: it is not important that these are “COVID-19” datasets. Suggest generalizing this section by reframing as “Biological interpretation of clinical metabolomics data”.

Response: Thank you for the suggestion. We have renamed the section based on your suggestion.

- Line 266: this should be the highlighted point of the manuscript! It is buried here in this section.

Response: Thank you. This is now one of the main points in the revised manuscript

- Line 276: highlight the automated parameter tuning without comparison to other tools. You can just highlight this as an advantage of your tool.

Response: Yes, we have emphasized this point in our discussion

- In general, the # of features detected is not diagnostic of “correctness” – especially in a complex sample where we do not know what “correct” is. This is also true for pathway prediction (line 291) – just predicting “more” pathways does not mean better.

Response: We have de-emphasized the focus on the number of features and eliminated comparisons to other tools. Instead, we now briefly describe the results obtained from MetaboAnalystR, including detected features and predicted pathways. It is important to note that having "more" pathways does not necessarily equate to better results or findings. Therefore, in the biological interpretation subsection, our focus is on whether biologically meaningful pathways have been increased. In this context, we have observed that the newly identified pathways are

relevant to the disease based on previous studies. In presenting these findings, we have endeavored to be correct and concise.

- Line 301: change “COVID-19” to “clinical metabolomics” (or similar).

Response: Thank you, we have made the updates.

- I do not feel that the conclusions in the discussion are supported by what is presented. For example, line 331 – you cannot claim that you have more “accurate functional enrichment analysis”. Again, it would be more appropriate to focus on the capabilities of your tool not the comparison to other tools.

Response: We fully agree with your advice regarding the emphasis of this manuscript. The discussion section has been rewritten accordingly, with all non-related content or unnecessary results removed. The description of “more accurate” functional analysis results has been revised only in the context of incorporating MS2 data compared to the results based on MS1 alone.

In the discussion, we concentrated on the innovative parts of the current study, as well as the role of MetaboAnalystR as a unified workflow for metabolomics field. In addition, we have also mentioned our future plan and coming update.

Reviewer #2 (Remarks to the Author):

Response: Thanks. We have systematically rewritten the manuscript based on reviewers’ comments to systematically improve the quality.

Reviewer #3 (Remarks to the Author):

This manuscript is a comprehensive report on the performance of the MetaboAnalystR 4. The amount of conducted studies is overwhelming and is also impressive. The authors should have spent much time and effort to produce this much results and assessments. I really admire this thorough effort to improve the efficiency of metabolomics research. This said, my overall impression of this manuscript is that the question on novelty: what is the strong benefit or the advancement in terms of science? Maybe the power of integration is one answer, but the current manuscript fails to address the true novelty of the contribution because the performance is rather competing than to be called excelling. The novelty in my understanding is the peak-wise introduction of the elastic-net optimization, assessing as many as 110 parameter combinations for every detected peak (Page 26). Honestly, I cannot believe that this much computation can compete with existing software platforms in time, but anyway, this is what is described. So if the authors wish to elaborate on the scientific novelty,

this optimization part should be the focal point. My doubt is that the elastic-net may be unnecessary; the performance of identification may be equivalent by just using LASSO.

Response: Thank you for your comprehensive review and comments on this manuscript. We have systematically rewritten the entire manuscript based on the reviewer's feedback and our recent updates following the initial submission. Instead of superficially comparing with other tools, we have focused on highlighting the innovative aspects and the unique role of MetaboAnalystR as a unified workflow for global metabolomics data processing and analysis.

Regarding the auto-optimized workflow, which evaluates 110 parameter combinations, we fully understand the concerns and would like clarify several points:

1. Auto-optimization is only applied to the actually convolved DDA spectra, which typically constitute less than 50% of the data.
2. Some convolved DDA spectra may not have any reference spectra from the database, resulting in fewer spectra needing optimization.
3. The auto-optimization was programmed in a C++ environment, ensuring the computational efficiency of this speed-determining step.

My suggestion for the authors is that they should focus more on such methodological improvement, rather than a superficial competition between software platforms. In my opinion, comparison with MS-DIAL/MS-FINDER or MZmine/SIRIUS is futile because the software performance hugely depends on parameter optimization (of course!), and the default setting is not good at all. In other words, we should NOT compete in performance but in rigorous methodology / theory. Since the current manuscript leans too much toward beating other software platforms, the manuscript became unnecessarily lengthy and descriptive on actual performance. My preference is to bring the Method part more to the front-end and push aside the actual performances in supplements.

Response: Thanks for this advice. We fully agree with you. In this revision, we have minimized the comparison to other tools. Most comparison results have been moved to supplements as a reference for users/readers. The results section has been tremendously simplified to avoid any confusing results or content. Based on your advice, we have brought more methodological descriptions to each of beginning of the results sections to ensure users/readers could easily follow and understand the innovative parts and the general design of MetaboAnalystR.

Major recommendation:

1. Rather than focusing on the relative performance of existing software platforms, the authors need to focus on the theoretical / methodological advantage of this contribution.

Response: Yes, Thanks for this advice. We have minimized the comparison to other tools in the manuscript since they are evaluated only based on their default settings. In contrast, in the present study, our focus is the unified metabolomics workflow for global metabolomics data processing. Therefore, this revision mainly emphasizes the innovative parts as well as the key role of a unified workflow for metabolomics data processing.

2. This means that description on the neutral-loss searching and the replicate treatment (Page 27-

28) needs elaboration, because the current explanation is hard to understand (e.g. what is the NL reference library and how multiple matches are handled).

Response: Thanks for your advice. We have added some explanation on neutral loss reference library as well as how to deal with the multiple matches in the “Neutral loss searching” and “Result export” subsections.

3. The decoy library is not really a decoy, since it’s a standard mixture. A more detailed explanation is needed to accurately evaluate the decoy performance, since this work includes not only small metabolites but lipids and exposome metabolites.

Response: Thank you for your comment. Following your suggestion, we have provided more details on the decoy-based evaluation in the Method section.

The decoy spectra are derived from a standard mixture. However, unlike the real samples, the decoy spectra are generated by introducing synthetic decoy MS2 spectra and replacing the actual MS2 spectra. After processing, the standard mixtures become no longer "valid" standard mixtures. The MS2 spectra within the decoy spectra data are considered "true negatives." If they are still reported as their original identities by data processing tools, these results should be considered "false positives." By employing this approach, we can effectively evaluate the performance of tools in identifying false positives.

Regarding the different compositions within the standard mixture, indeed, there are multiple components such as metabolites, lipids, and even exposome metabolites. However, the decoy-based testing was conducted based on the complete MS2 database for MetaboAnalystR, while other tools also utilized their comprehensive databases. In such cases, all compounds were covered by their respective databases. We hope this clarifies any confusion.

4. Availability of blood sample data and software R code is welcome but researchers also need data. Is it possible to share curated data?

Response: Yes. All raw spectral data, software and data analysis code has been included in Data Availability section or Code Availability section. I understand the “curated data” you are referring to is the curated MS2 reference database. We have added the link of these curated spectral databases from the Data Availability section.

5. Description of the mummichog algorithm is missing. Please describe at least its overview in the Method section (P32 is not enough at all).

Response: We have added an overview on mummichog algorithm in the subsection, “*Integrating of LC-MS data processing results to functional insights*”, to help readers and users quickly understand the background and basic principles of the algorithm.

6. Please add availability of curated spectral data.

Response: Thank you. All curated spectral data has been declared in the Data Availability section.

Reviewers' Comments:

Reviewer #1:

Remarks to the Author:

In this work, the authors detail the development of MetaboAnalystR 4.0, a novel end to end data processing software for untargeted metabolomics data. This is an incredibly exciting frontier for the field of metabolomics. The authors have integrated feedback from reviewers which has significantly improved the manuscript.

Revisions and Comments:

1. Generally, the readability of the manuscript could be improved to fix grammatical errors. It is recommended that the authors utilize an English language editing service prior to final submission. Some examples are noted below:

a. Line 21. Should be "workflows" or "an easy-to-use, comprehensive computational workflow..."

b. Line 26, "spectra data" should be "spectra"

c. Abstract line 27 "by integrating multiple information from spectral annotation" is an ambiguous statement.

d. Line 36: "Metabolomics involves the comprehensive identification and quantification of small compounds"

e. Line 99: "MetaboAnalystR features an auto-optimized LC-MS1 spectra pre-processing module since version 3.0" should be re-written.

f. Line 179: "generated a whole blood exposomics"

g. Line 189-190: "A clustering heatmap with was used.."

h. Line 194: "metabolic" should be "metabolites"

i. Line 207: "from the Mass Spectrometry Metabolite Library". I would also suggest a re-introduction of what the SM abbreviation stands for.

j. Line 369-370: "No QC samples were prepared in this case study"

k. Line 421: "North America"

l. Line 547: "It is a common practice for MS2 data acquisition with multiple replicates." should be rewritten

m. Line 581: should be "characterizing"

2. The abstract could more explicitly state how tools exist for the individual parts of the untargeted metabolomics data processing pipeline and the innovation of MetaboAnalystR 4.0 is the end-to-end processing of data all within the MetaboAnalystR 4.0 tool.

3. Throughout the manuscript, the authors note how MetaboAnalystR 4.0 can be implemented with raw data. It should be specified if this raw data must first be converted to a universal file type (i.e. mzML), or if vendor-specific data files can be used.

4. Paragraph 2 of the introduction lacks a connection of how DDA and DIA data relate to the development of MetaboAnalystR 4.0. Perhaps to bring out the unique challenges of DDA and DIA the MetaboAnalystR 4.0 is addressing, this paragraph could be re-structured to highlight how these methods of generating tandem MS spectra have unique challenges that require unique solutions. Most metabolomic processing tools focus on either DDA or DIA but not both. This is another benefit of MetaboAnalystR 4.0 that is not as highlighted as it should be.

5. There appears to be no reference to Figure 2 in the main manuscript text.

6. From the main manuscript text, it is unclear how alignment of features across experiments is performed. Additionally, how are situations handled where a metabolite is annotated in one experiment group but not another?

7. In the SM dataset results section (Line 217), it is unclear what the increased number of compounds being correctly identified is in reference to. These results could be better set up by clarifying the reference being used. Additionally, the manuscript could benefit from a definition of what must be true for a compound to be considered detected. Are data assessed based on precursor observation alone or is other data required?

8. The COVID studies referenced in line 243 should be cited.

9. Supplemental table S2 and S3 show a summary of true positives obtained with various MS processing tools. Were false positive rates also considered? It appears that generally, MetaboAnalystR 4.0 is observing more standard matches and more compound annotations in complex samples. The false positive rate being similar across tools would help validate these annotations as real compounds.

10. For the lipidomics database, it is common that lipids are often over annotated to include unknown information (i.e. double bond position and orientation). Have the authors validated identifications in databases and their agreement with the data generated from the LC-MSMS experiments that users will submit to MetaboAnalystR 4.0?

Reviewer #2:

Remarks to the Author:

Reviewer #3:

Remarks to the Author:

The authors have revised the manuscript fully and it was improved much. My comments were well addressed in the revision. Now I could understand the protocols. Putting all comparisons in the supplementary part is also good. The results obtained are weak, not showing any discovery or new findings, but my understanding is that these are only cases studies to exemplify the ability of the system.

Having read through the revision, my minor recommendations are as follows.

1. Please do not use "auto-optimized". From the beginning the authors emphasize the "auto-optimization", but as far as I see the methods section, no particular optimization is performed. It is a standard, ad-hoc method for peak detection and identification. The process should be therefore "automated" rather than "optimized".

2. Describe more on your online web server. MetaboAnalyst prepares a very good website with detailed tutorials. I suppose the current version 6.0 is based on MetaboAnalystR 4.0, but somehow this relationship is not well documented in the manuscript. (Do authors present the website in a separate journal? Then a more recent paper should be cited.)

REVIEWERS' COMMENTS

Reviewer #1 (Remarks to the Author):

Revisions and Comments:

1. Generally, the readability of the manuscript could be improved to fix grammatical errors. It is recommended that the authors utilize an English language editing service prior to final submission. Some examples are noted below:

- a. Line 21. Should be “workflows” or “an easy-to-use, comprehensive computational workflow...”
- b. Line 26, “spectra data” should be “spectra”
- c. Abstract line 27 “by integrating multiple information from spectral annotation” is an ambiguous statement.
- d. Line 36: “Metabolomics involves the comprehensive identification and quantification of small compounds”
- e. Line 99: “MetaboAnalystR features an auto-optimized LC-MS1 spectra pre-processing module since version 3.0” should be re-written.
- f. Line 179: “generated a whole blood exposomics”
- g. Line 189-190: “A clustering heatmap with was used..”
- h. Line 194: “metabolic” should be “metabolites”
- i. Line 207: “from the Mass Spectrometry Metabolite Library”. I would also suggest a re-introduction of what the SM abbreviation stands for.
- j. Line 369-370: “No QC samples were prepared in this case study”
- k. Line 421: “North America”
- l. Line 547: “It is a common practice for MS2 data acquisition with multiple replicates.” should be rewritten
- m. Line 581: should be “characterizing”

Response: Thanks for these corrections. We have corrected all these points based on your suggestions.

2. The abstract could more explicitly state how tools exist for the individual parts of the untargeted metabolomics data processing pipeline and the innovation of MetaboAnalystR 4.0 is the end-to-end processing of data all within the MetaboAnalystR 4.0 tool.

Response: Thank you. We have updated the Abstract section based on your advice.

3. Throughout the manuscript, the authors note how MetaboAnalystR 4.0 can be implemented with raw data. It should be specified if this raw data must first be converted to a universal file type (i.e. mzML), or if vendor-specific data files can be used.

Response: MetaboAnalystR 4.0 accepts open formats (mzML, mzXML, mzData, netCDF and mgf) at the current stage. We have clarified this point in the “General workflow” section of the manuscript based on your suggestion.

4. Paragraph 2 of the introduction lacks a connection of how DDA and DIA data relate to the development of MetaboAnalystR 4.0. Perhaps to bring out the unique challenges of DDA and DIA the MetaboAnalystR 4.0 is addressing, this paragraph could be re-structured to highlight how these methods of generating tandem MS spectra have unique challenges that require unique solutions. Most metabolomic processing tools focus on either DDA or DIA but not both. This is another benefit of MetaboAnalystR 4.0 that is not as highlighted as it should be.

Response: Thanks for the. We have updated both Paragraph 2 and 3 based on your advice. We have emphasized that not only a general or unified workflow is missing, but also there are significant computational bottlenecks with the current implementations for both DDA and DIA. The development of MetaboAnalystR 4.0 is to address the challenges and unify the workflow. We have also highlighted that MetaboAnalystR could process both DDA and DIA, while most of others cannot in paragraph 2.

5. There appears to be no reference to Figure 2 in the main manuscript text.

Response: Thanks for your reviewing. Figure 2 was referred to in the second paragraph of the subsection, “*LC-MS spectra processing*” of the results section.

6. From the main manuscript text, it is unclear how alignment of features across experiments is performed. Additionally, how are situations handled where a metabolite is annotated in one experiment group but not another?

Response: Thanks for the comment. Since the alignment of LC-MS1 features are not the innovative part in the version (4.0), this is not mentioned. The key issue related to the alignment of LC-MS1 features across different experiments is the retention time drift which has been addressed in MetaboAnalystR 3.0. Briefly,

- MetaboAnalystR uses the same approach as in XCMS for general peak grouping;
- For different batches, MetaboAnalystR uses QC-RLSC method for batch effect correction;

For MS2 spectra, we have developed a method to merge and consensus the spectra from multiple samples (replicates of MS2 spectra) and perform compound identification based on the consensus spectrum. In this situation, there is only one consensus spectrum existing for compound identification. In addition, based on the spectra consensus rule, the low-quality spectrum/fragmentation will be removed at the step to reduce potential matching error.

7. In the SM dataset results section (Line 217), it is unclear what the increased number of

compounds being correctly identified is in reference to. These results could be better set up by clarifying the reference being used. Additionally, the manuscript could benefit from a definition of what must be true for a compound to be considered detected. Are data assessed based on precursor observation alone or is other data required?

Response: Thank you for pointing this out. The information of the standards in each SM dataset includes retention time, m/z of precursor, and the exact chemical identity. Therefore, we utilized the corresponding information of these compounds as the reference (i.e. ground truth) to evaluate the accuracy of compound identification. We have clarified this point based on your comments in the previous paragraph of the comparison content. We have also added the definition of compound identification criteria in the Method section, *MetaboAnalystR usage*.

8. The COVID studies referenced in line 243 should be cited.

Response: We have added references of the two COVID studies to the manuscript.

9. Supplemental table S2 and S3 show a summary of true positives obtained with various MS processing tools. Were false positive rates also considered? It appears that generally, MetaboAnalystR 4.0 is observing more standard matches and more compound annotations in complex samples. The false positive rate being similar across tools would help validate these annotations as real compounds.

Response: Thank you for the question. We indeed performed assessment on the false positive rate of MeataboAnalystR 4.0 by using a series of decoy spectra samples. The description was provided in the third paragraph of the section, "Performance of LC-MS2 spectral processing".

10. For the lipidomics database, it is common that lipids are often over annotated to include unknown information (i.e. double bond position and orientation). Have the authors validated identifications in databases and their agreement with the data generated from the LC-MSMS experiments that users will submit to MetaboAnalystR 4.0?

Response: Thanks for these comments. We totally agree with you. The lipidomics database actually includes a lot of over-annotated information on the complex chemical structure. In this study, we did not perform specific validation on these databases, but we do encourage users to use their own validated in-house database by formatting it into the correct open-source format. In the future, we will further curate and validate the database, especially the ones predicted *in silico*.

Reviewer #3 (Remarks to the Author):

1. Please do not use "auto-optimized". From the beginning the authors emphasize the "auto-optimization", but as far as I see the methods section, no particular optimization is performed. It is

a standard, ad-hoc method for peak detection and identification. The process should be therefore "automated" rather than "optimized".

Response: Thanks for this comment. Automated means chaining different steps together, which is relatively straightforward for a command line tool such as R or python packages. In MetaboAnalystR, we have implemented auto-optimization methods for both LC-MS1 and MS2 spectra processing. For LC-MS1, we utilized a regions-of-interests (ROI)-based method to automatically optimize the parameters for feature detection. The detailed algorithm was published in the previous release of MetaboAnalystR 3.0. For DDA spectra deconvolution in LC-MS2, we have also implemented an optimization to optimize the best combination of α and λ values for the elastic-net model. This optimization is also performed automatically, we think "auto-optimized" is more accurate in this case.

2. Describe more on your online web server. MetaboAnalyst prepares a very good website with detailed tutorials. I suppose the current version 6.0 is based on MetaboAnalystR 4.0, but somehow this relationship is not well documented in the manuscript. (Do authors present the website in a separate journal? Then a more recent paper should be cited.)

Response: Thank you for the question. The MetaboAnalyst 6.0 web server is based on MetaboAnalystR 4.0. The MetaboAnalyst 6.0 has been accepted by the journal of Nucleic Acid Research in its 2024 web server issue. We have added the reference.

3. Please double check http links. At least the following links are NOT accurate. I could not verify links in the supplementary files, so please check.

- Library on the web server: <https://www.metaboanalyst.ca/docs/databases.xhtml> (Database.xhtml is the correct address.)

Response: Thank you for pointing this out. The URL has been updated recently. We have corrected it in the manuscript.

- Curatr website of EMBL: <https://curatr.mcf.embl.de/> (HTTP is the correct protocol.)

Response: Thank you for the comment. The certificate of this EMBL website has expired recently. We will notify the developer to update their certificate.

- Zhiqiang-Pang code on GitHub: https://github.com/zhiqiang-PANG/MetabR4_scripts (No idea for the correct address)

Response: It was not open to the public before. We have open it the public for users to reuse the code.